# Examining Exposure Fires from the United States National Fire Incident Reporting System between 2002 and 2020

Derek J. McNamara [1,*] and William E. Mell [2]

1 Geospatial Measurement Solutions, LLC, Sisters, OR 97759, USA
2 United States Forest Service, Washington, DC 20250, USA; william.mell@usda.gov
* Correspondence: dmgeo@gmsgis.com

**Abstract:** Fires resulting from antecedent fires, known as exposure fires, can manifest across diverse environments, including suburban, urban, and rural areas. Notably, exposure fires represented by structure-destroying fires within the wildland–urban interface (WUI) can extend into non-WUI suburban and urban regions, presenting significant challenges. Leveraging data from the United States National Fire Incident Reporting System (NFIRS) spanning 2002 to 2020, this study investigates 131,739 exposure fire incidents impacting 348,089 features (incidents). We analyze reported economic costs, affected feature types, and property utilization patterns for these exposure fires. We also compare these exposure fires to information documented in other databases. Finally, we examine structure separation distance at residential dwellings and describe ignition pathways for selected fires. Reported property losses for some fire incidents amounted to USD 5,647,121,172, with content losses totaling USD 1,777,345,793. Prominent fire incident categories include buildings, vehicles, and natural vegetation fires, predominantly occurring in residential, outdoor, and storage areas. While the NFIRS lacked information on most major structure-destroying WUI fires, highlighting this analysis's lack of statistical representation, it did provide insights into less extensive exposure fires, both WUI and non-WUI, unrecorded elsewhere. Our study reveals significant distinctions in the distribution of separation distances between damaged-to-damaged structures (average separation of 6.5 m) and damaged-to-not-damaged structures (average separation of 18.1 m). Notably, 84% of the incidents in exposure fires involved fire suppression defensive actions. These defensive actions contributed to the differences in structure separation distance distributions, highlighting the often-neglected role of these measures in assessing structure responses during WUI fires. We examined ignition pathways at select exposure fires, highlighting some common features involved in fire spread and challenges in documenting these pathways. Finally, we propose a set of idealized attributes for documenting exposure fires, accentuating the inherent difficulties in collecting such data across expansive geographical areas, particularly when striving for statistical representation. Our findings yield valuable insights into the multifaceted nature of exposure fires, informing future research and database development to aid in mitigating their impact on vulnerable communities.

**Keywords:** National Fire Incident Reporting System (NFIRS); wildland–urban interface (WUI); exposure; exposure fire; structure separation distance; ignition pathways

## 1. Introduction

The largest fires in terms of structure destruction have occurred in recent years (e.g., 2023 Hawaii Lahaina Fire, 2021 Colorado Marshall Fire, 2020 Oregon Almeda Drive Fire, 2018 California Camp Fire, 2017 California Tubbs Fire, and 2016 Tennessee Chimney Tops 2 Fire). Structure destruction at these fires, while initiated by wildland fire, was primarily a result of urban conflagrations [1]. Predictions of wildfires increasing due to climate change [2,3], accelerating housing growth [4], increases in urban fires due to rising temperatures [5], and escalating demands on firefighting resources [6] emphasize the

importance of examining structure conflagrations, and other exposure fires, in wildland–urban interface (WUI) and non-WUI environments.

The United States (U.S.) National Fire Incident Reporting System (NFIRS) [7] defines exposure fires as "fires resulting from another fire outside that building, structure, or vehicle, or a fire that extends to an outside property from a building, structure, or vehicle [7]". The originating fire is termed a source fire and includes wildland fires [8]. We refer to the combination of the source and the subsequent fire incident(s) as exposure fires, of which WUI and urban fires are a growing concern [9,10].

The WUI occurs where wildlands and human developments intersect. Fires in the WUI can also encroach into suburban and urban areas not classified as WUI [1], resulting in significant structure conflagrations, such as in the 2017 California Tubbs Fire Coffey Park neighborhood. As resources become limited (e.g., gasoline and water in some locations) and meteorological conditions (e.g., drought and wind speed) become extreme [3], suppressing exposure fires in the WUI and elsewhere can become more challenging, resulting in significant destruction, evacuations, and community disruption.

WUI fires occur when multiple structure ignitions from wildland fires overwhelm existing suppression capabilities [9]. Containing these ignitions is challenging. Also, these ignitions can spread fire by heat fluxes from flames and embers to other structures (and features between structures, such as fences and vegetation). This fire spread can continue unabated until there is a change in conditions. For example, changes in wind, humidity, structure separation distance (SSD), human intervention, land use/land cover (LULC), building materials, or other factors might reduce the magnitude or effects of heat fluxes from flames and embers to the built environment and enable fire containment. Numerous studies (e.g., [11–14]) have attempted to unravel these complexities to determine the primary causes of structure destruction.

Still, current research has not quantified the exact contribution of different ignition mechanisms (heat, flames, embers) or the predominant source of heat flux (radiations, conduction, convection) leading to structure destruction over significant spatiotemporal extents. Furthermore, there is limited reliable information regarding the conditions leading to fire spread in exposure fires due to confounding factors in post-fire environments. For example, McNamara and Mell [15] and McNamara, Mell, and Maranghides [16] showed a dependence on the stopping of fire spread and indicators of defensive actions (e.g., evidence of water suppression) from eyewitness discussions and remote sensing data.

However, researchers have considered confounding factors in assessing structure response to fire, such as defensive actions, inconsistently. For example, Knapp et al. [12] found few defensive actions at the 2018 California Camp Fire, excluding them as a predictor variable in their assessment. Conversely, Troy et al. [13] found defensive actions to be the most critical predictor variable in their analysis. A growing body of research highlights the importance of defensive actions in assessing structure response at WUI fires (e.g., [15–17]).

Compounding the difficulties in understanding the WUI problem is a lack of national databases representing the results of post-fire assessments and the full extent of structure destruction from exposure fires. However, NFIRS provides information on structures destroyed and damaged by fires involving numerous human-made and natural features. NFIRS is a voluntary system designed to report all fires and other incidents to which fire departments respond. Although NFIRS-reporting fire departments cover 71% to 83% of the U.S. population [7], research has shown that the NFIRS only contains about 44% of all U.S fires [18].

Additionally, Butry and Thomas [19] found an underreporting of NFRIS-documented WUI fires in California between 2004 and 2013. However, they did not examine all reported exposure fires during this time. Existing examination of NFIRS exposure fires is currently limited. The U.S. Federal Emergency Management Agency [8] analyzed exposure fires from the 2004 NFIRS database. The U.S. Fire Administration (USFA) [20] examined NFIRS fires in the California WUI, including exposure fires, between 2009 and 2011. Finally, the U.S. Fire Administration (USFA) examined six large WUI fires in the U.S. [21], including some

incidents reported as exposure fires. While the above studies highlight the underreporting of WUI fires in the NFIRS to a limited extent, no study has examined NFIRS source fires to a significant spatiotemporal extent.

Identifying incidents occurring within known wildfire boundaries and the dates those wildfires occurred can provide information on NFIRS-reported structures affected by wildfires [21]. This approach requires ancillary data (e.g., wildland fire perimeters) to identify affected structures. Additionally, wildland fire incidents can have information recorded in the NFIRS Wildland Fire Module, containing the number of structures ignited and threatened by the wildland fire. However, this approach does not document the locations of affected buildings, the type of affected building (e.g., outbuilding), or other affected features such as vehicles. Also, fire departments are not required to complete the NFIRS wildland module, resulting in limited data in this module [19].

Studying NFIRS-reported exposure fires over a significant spatiotemporal extent might aid in understanding conditions that result in exposure fires, such as WUI fires that produce significant structure conflagrations. Consequently, in this study, we utilize NFIRS data covering U.S. and Puerto Rico reported between 2002 and 2020 to address the following questions:

1.  What are the characteristics of exposure fires reported in the NFIRS regarding losses, affected features, defensive actions, heat sources, and locations?
2.  How do NFIRS-reported large exposure fires (greater than ten features affected) compare to reported damage in other databases?
3.  Can NFIRS exposure fires be used to assess the effects of structure separation distance (SSD) on structure-to-structure fire spread?
4.  Can NFIRS exposure fires be used to assess ignition pathways present in these fires?

We also compare results from our examination of SSD and defensive actions in the NFIRS to conditions at the 2018 California Woolsey Fire (Woolsey Fire) [11], which is not reported in the NFIRS, to highlight poorly founded assumptions in many WUI post-fire assessments. Addressing the above questions also provides insights into the characteristics needed in a database of exposure fires to aid the study of the growing WUI problem and other exposure fires.

## 2. Materials and Methods

The NFIRS is a voluntary database system designed to document the location, mitigation resources, losses, and other information for fire, hazardous material, explosion with no fire, and emergency medical service incidents responded to by fire departments [7]. The NFIRS contains eighteen tables organized into a relational database schema [22]. This study focuses on exposure fires documented in the BasicIncident, FireIncident, and IncidentAddress tables [22].

The NFIRS records each fire incident (incident) in an exposure fire as a separate incident or record in the BasicIncident table. Each incident in an exposure fire should have a corresponding record in the FireIncident and IncidentAddress tables. Each incident in an exposure fire should have the same attribute values for state (the U.S. state where the incident occurred), fdid (a unique identifier for the fire department), date (the date of the incident), and inc_no (a unique identifier for the incident) in each of the three tables examined. The first incident or initially ignited feature should have a zero value for the exp_no attribute and is called a source fire [8]. Subsequent incidents have values increased by one for each incident (e.g., structure or vegetation) in the exposure fire. An example of an NFIRS-reported exposure fire with three incidents is shown in Figure 1.

In this study, NFIRS data between 2002 and 2020 were loaded to a PostgreSQL 11 database containing the NFIRS database schema [22]. This database contained 22,575,246 records in the BasicIncident table, with 20,411,306 fire incidents. The structured query language (SQL) statements used to extract exposure fires are listed in Appendix A.

The data produced from these queries were exported from the PostgreSQL database and imported into an ESRI™ File Geodatabase. Those exposure fire incidents in the

exported dataset with sufficient location information (e.g., incidents with a specific address, including the house number) were then geolocated in ArcGIS Pro™ and loaded to the same File Geodatabase containing the tabular data.

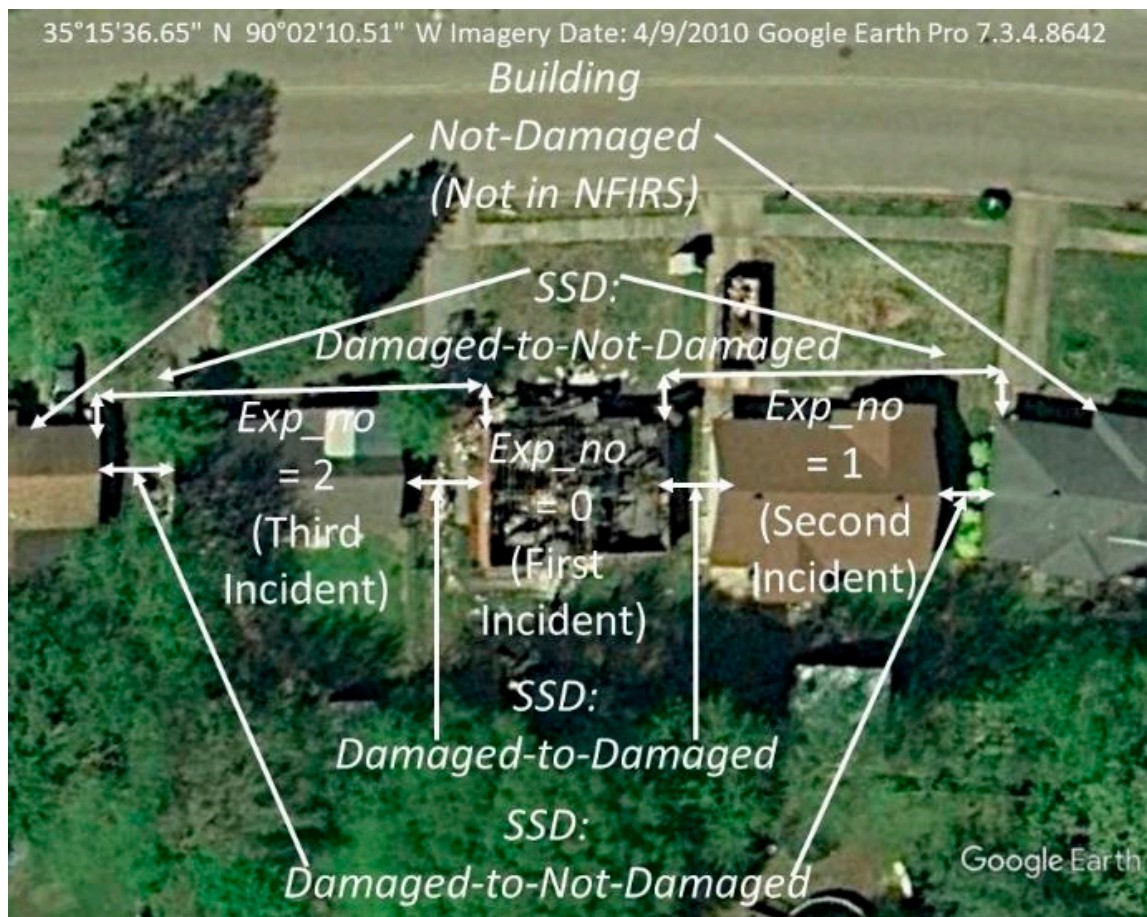

**Figure 1.** An example of an NFIRS-reported exposure fire is portrayed in Google Earth™ Imagery. Exp_no 0 was destroyed, and exp_no 1 and 2 were damaged, presumably from exp_no 0. Examples of SSD calculations performed in this study between damaged-to-damaged and damaged-to-not-damaged structures for this exposure fire are also shown.

The tabular dataset produced from the exposure fire queries (Appendix A) was used to provide information about the reported characteristics of these exposure fires. This summary included the number of exposure fires, the total property loss and content loss reported, defensive action quantities by type, heat sources by type, and information about the incident type and property use. Also, geolocated exposure fire incidents were examined by 2010 WUI type [23] for incidents before 2010 and 2020 WUI type [23] for incidents occurring on or after 2010.

Additionally, exposure fires with greater than 20 incidents were examined. The type of fire (e.g., WUI, wildlands, suburban, urban, apartment only) was described based on examining the location in Google Earth™. Wildland fires contained only vegetative fire incidents. WUI fires contained at least one vegetative fire incident with a human-made fire incident type or, in some cases, contained no vegetative fire incidents but were known to be started by wildland fires (e.g., the 2016 Montana Roaring Lion Fire). Urban fires occurred in areas with limited to no vegetation. Suburban fires occurred in residential areas, and suburban/urban fires occurred in areas with a mix of suburban and urban areas as interpreted by overhead images in Google Earth Pro™.

For these larger exposure fires, it was determined if internet searchers (e.g., CAL FIRE incident reports) contained information on structures affected by the fire. If there was no

publicly available information on structures affected, data from the SIT-209 report [24], which collects and stores summary information (e.g., structures destroyed) on significant fire incidents, were examined. Finally, at the 2011 Texas Tanglewood Fire, a further comparison of the NFIRS-reported incidents against those documented from detailed post-fire ground assessments [25,26] was conducted to examine the potential underreporting.

Following the above, SSD was assessed for select NFIRS exposure fires. The Microsoft (MS™) dataset [27] of building footprints (footprints) was used to calculate SSD between damaged-to-not-damaged and damaged-to-damaged residential structures. The SSD between the identified building footprints (representing incidents in exposure fires identified as described below) was calculated (Figure 1) using a custom Python script in ArcGIS™ Pro 2.9. This script utilized the Near tool in ArcGIS™ Pro 2.9. The distance between footprints (polygons) is calculated as the closest distance between two polygon boundaries. If polygons overlap or touch, the distance is recorded as 0.

This SSD assessment required additional filters beyond those described in Appendix A. First, the dataset of exposure fires was filtered for single or multifamily structures (i.e., inc_type equal to 111 and prop_use equal to 419 or 429). Geocoding of NFIRS exposures is often by address with the point location occurring on the primary structure (e.g., home). Consequently, other incident types (e.g., vehicles) could not be precisely located beyond the primary structure location.

Selecting only those incidents with an inc_type equal to 111 (building fire, not confined) and prop_use equal to 419 or 429 (one- or two-family dwelling or multifamily dwelling, respectively) resulted in some exposure fires with only one incident (e.g., an exposure fire where a vehicle ignited a home). These exposure fires with only one incident (i.e., no residential structure-to-structure fire spread occurred) were removed. After applying this filter, incidents containing information to geocode the feature at the structure level were identified. A subsequent filter excluded exposure fires on multifamily residences in one structure (e.g., apartment fires in the same structure).

The next filter ensured that the remaining exposures were consecutive (e.g., single-family structure to single-family structure). Therefore, incidents where the exp_no was not consecutive were removed (e.g., a three-exposure fire with two structures having exp_no of zero and two, respectively, and the second exposure with an exp_no of one was a vehicle and not geocoded). The remaining exposure fires also required a footprint in the MS™ dataset [27], containing 129,591,852 footprints, coinciding with the geolocated incident. It was assumed that rebuilt destroyed structures would have a similar footprint design.

Because not all destroyed structures were rebuilt, the MS™ footprints might have missed the original structure, or the geocoded point did not fall within the structure. Therefore, another filter ensured that all remaining incidents were consecutive, as missed footprints might have resulted in exposure discontinuities. Finally, the MS™ footprints within 200 m of the filtered footprints representing damaged buildings were extracted to represent the not-damaged buildings. The reduction in exposure fires resulting from each filter is reported below.

The distributions of the two SSD calculations were compared visually using density plots. Also, the Kolmogorov–Smirnov test evaluated whether the two distributions come from the same population. Additionally, an examination like that described above for exposure fires with greater than 20 incidents occurred for those exposure fires used in the SSD assessment with equal to or greater than ten incidents.

Also, it is essential to note that our calculation of SSD is the minimum distance between footprints. In cases where parallel building walls exist, this distance likely captures SSD appropriately in characterizing the ignition hazard due to heat flux exposure from flames and hot gases. However, not all adjacent buildings have parallel walls, so the SSD measure utilized here might exaggerate the potential heat flux exposure from flames and hot gases.

Furthermore, errors in the spatial representations of footprints could result in erroneous SSD measurements. However, MS™ [27] assessed the quality of the footprints, finding an intersection over union (measuring overlap quality against labels) value of 0.86,

a shape distance value of 0.4 (similarity of polygon outline), and a dominant angle rotation error (polygon rotation deviation) of 2.5. Regardless, the assessment presented here is not intended to provide a quantitative measure of safe SSD; instead, it will only present general trends in the data and identify shortcomings and improvements to assessing SSD.

Ignition pathways (e.g., structure ignites a vehicle) are also examined. In theory, the order of values in the exp_no attribute represents the order of ignitions in exposure fires when there are two incidents in the exposure. The source fire ignited the feature in the second incident documented for these exposure fires.

These exposure fires are assessed for specific ignition pathways (e.g., vehicle to structure). It is more challenging to assess specific ignition pathways from exposure fires with three or more incidents due to ignitions not always being linear. For example, the building labeled as the first exposure (exp_no equals zero) in Figure 1 caused damage to the second (exp_no equals one) and third exposures (exp_no equals two). However, in some cases, the second exposure might ignite the third exposure. Without ancillary data (e.g., post-fire imagery, as shown in Figure 1), the ignition orders for exposure fires with more than two incidents cannot be determined.

Finally, comparisons between the NFIRS and Woolsey Fire SSD assessments [11] are made, highlighting similarities between conditions in large structure destroying WUI fires and NFIRS exposure fires and common poorly founded assumptions present in many post-fire WUI studies. These assessments facilitate identifying some ideal characteristics for databases attempting to understand exposure fire characteristics better.

## 3. Results

### 3.1. Data Reductions

The extracted exposure fires represent less than 2% (348,089 of 20,411,306) of the fire incidents reported by NFIRS between 2002 and 2020. We use all extracted exposure fires (131,739 with 348,089 incidents, as shown in Figure 2) to examine the characteristics of NFIRS-reported exposure fires. We portray the data reduction for each filter employed in Figure 2, including geolocation as the fourth filter to identify the number of incidents coded as single/multifamily residences. However, we could geolocate 67% (234,137 of 348,089) of all the exposure fire incidents.

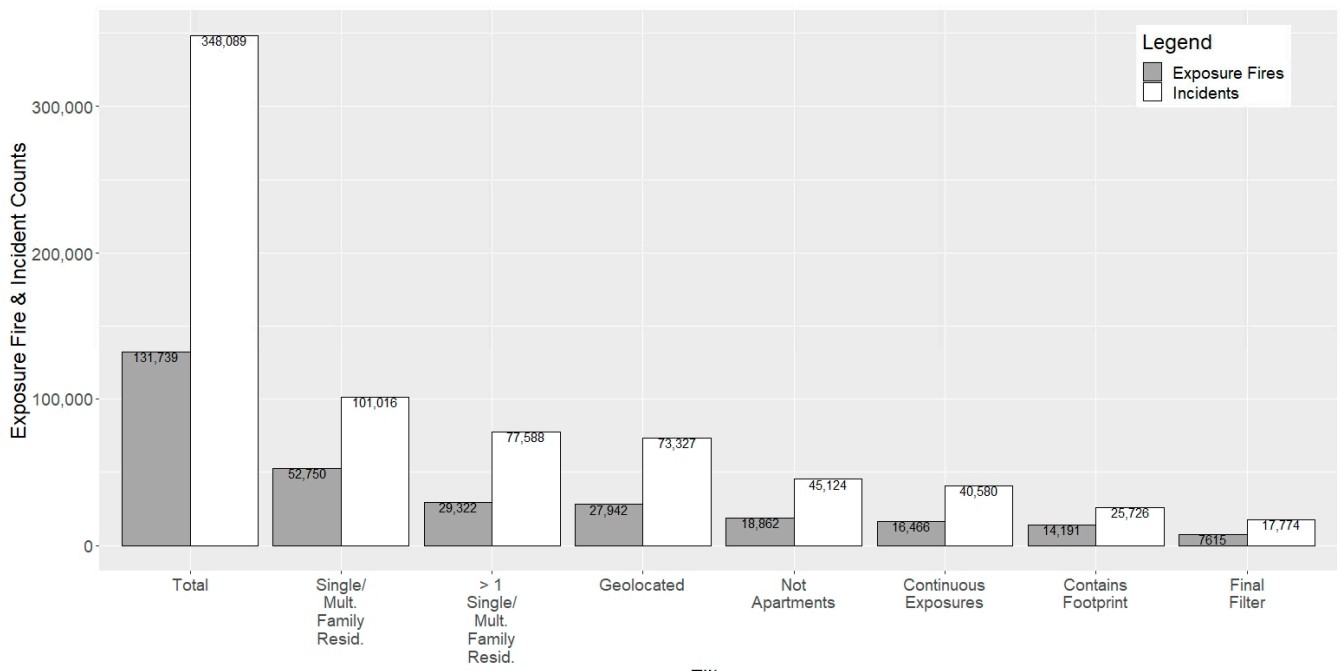

**Figure 2.** The data reduction results conducted to derive SSD in NFIRS exposure fires (2002 to 2020).

### 3.2. Distributions of Incidents, Incident Type, and Property Use

We show the number of incidents in each exposure fire in Figure 3. The majority of the 131,739 exposure fires, 99%, had ten or fewer incidents. Sixty-eight percent had two incidents (Figure 3). Only 26 exposure fires had more than 100 incidents (Figure 3). As discussed later, structure incidents in NFIRS-reported WUI fires do not coincide with other sources of information. However, 19,317 exposure fires with less than ten incidents (15%) contain a natural vegetation fire incident. An additional 1504 of these exposure fires contain a cultivated vegetation fire incident. These smaller WUI fires appear to represent fires not documented in other national databases.

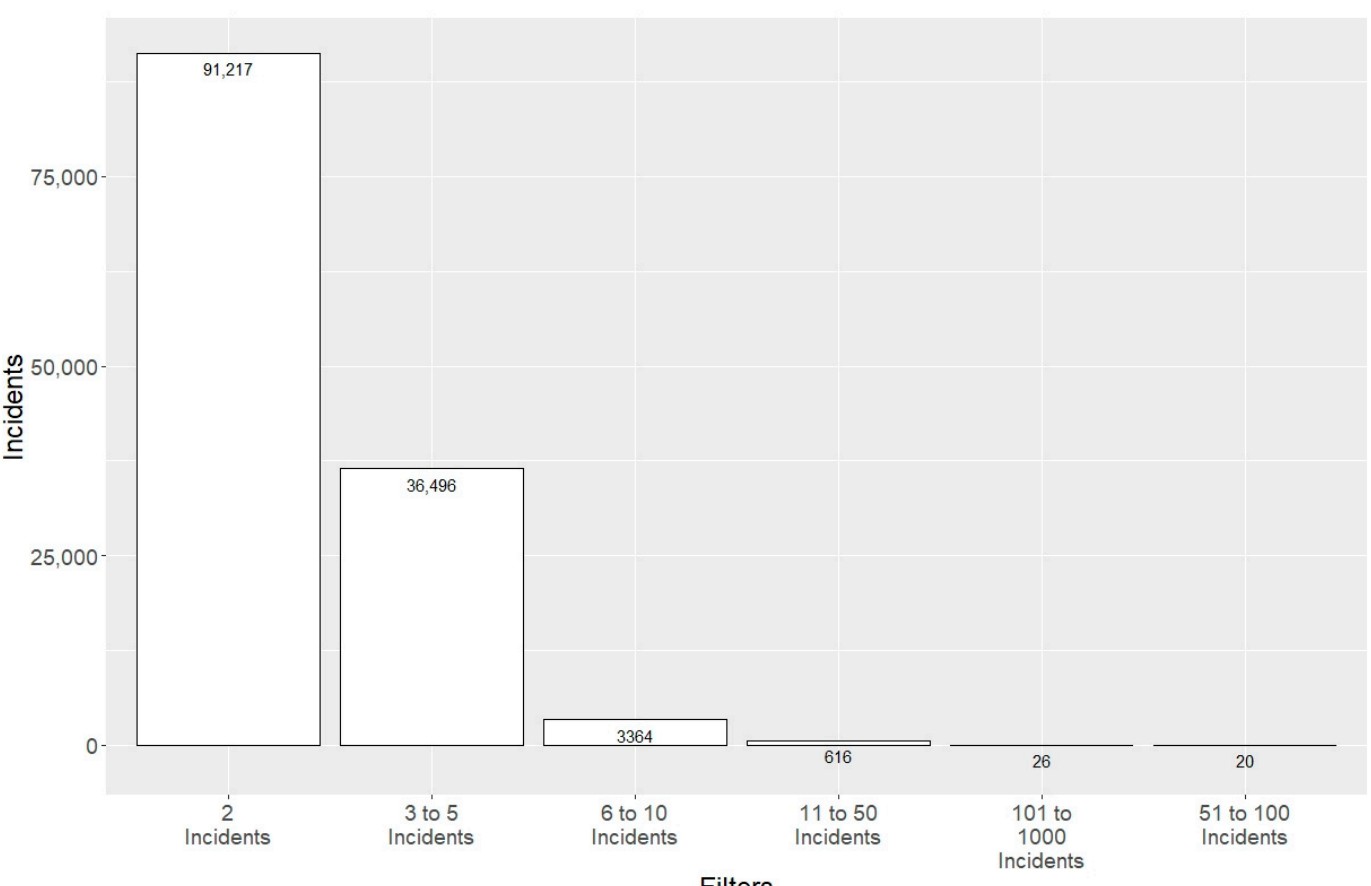

**Figure 3.** The number of exposure fires by the number of incidents per exposure fire in the NFIRS between 2002 and 2020.

We show the distribution of exposure incidents by incident type (inc_type field) in Figure 4a. Structures (buildings) represent the most numerous incident type for the exposure fires examined. Vehicles then natural fires follow structures in the number of incidents. Then, outside rubbish, undefined structures, special outside, mobile structures, other, structures other than buildings, confined structures, and cultivated vegetative.

There are 2773 (<1%) confined structure incidents in the exposure fires (Figure 4a). Confined fires are limited in extent, typically restricted to non-combustible containers. Therefore, they are not exposure fires and are data entry errors or improperly recorded as exposure fires.

We show the distribution of exposure incidents by property type (prop_type field) in Figure 4b. Forty-two percent of the incidents occurred on residential properties. Twenty-five percent occurred on outside or special use properties, followed by storage, other, and mercantile or business properties. The remainder of the incidents occurred on assembly, industry, manufacturing, educational, or health care or detention properties. Finally, there were 5559 (2%) incidents where the fire department did not record property use.

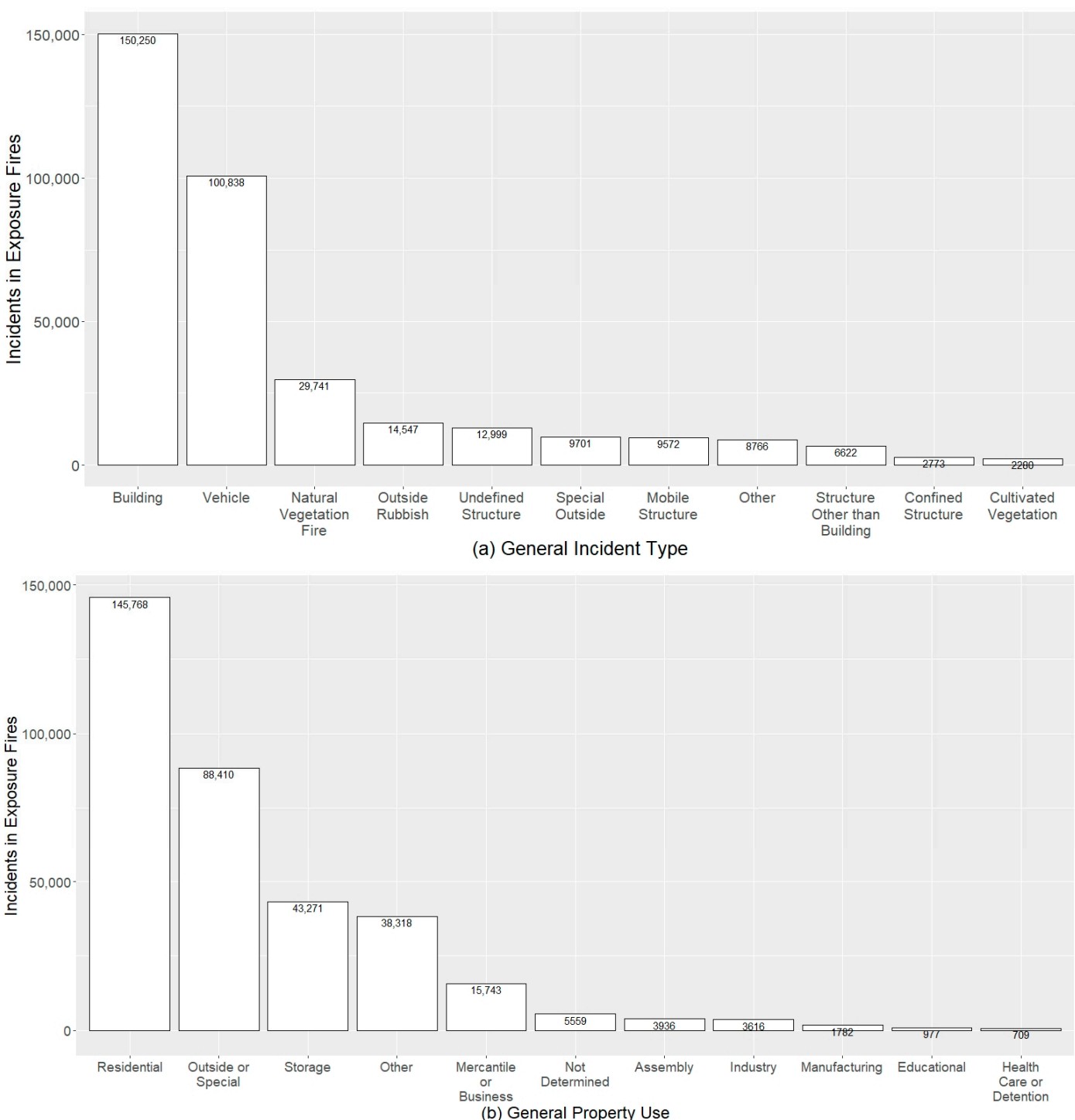

**Figure 4.** Distributions of the general incident and property use types for NFIRS exposure fires between 2002 and 2020. (**a**) Incident type. (**b**) Property use.

*3.3. Property and Content Losses*

Next, we describe the property and content losses for NFIRS-reported exposure fires by the number of incidents per U.S. dollar category and the total dollars reported (Figure 5). Many incidents had blank or "−99" values for the property (30%) and content (70%) loss (Figure 5a). Values of "−99" indicate that there was no significant loss. Furthermore, there are some extreme outliers. For example, three incidents have a total property loss of USD 547,401,649 (Figure 5a).

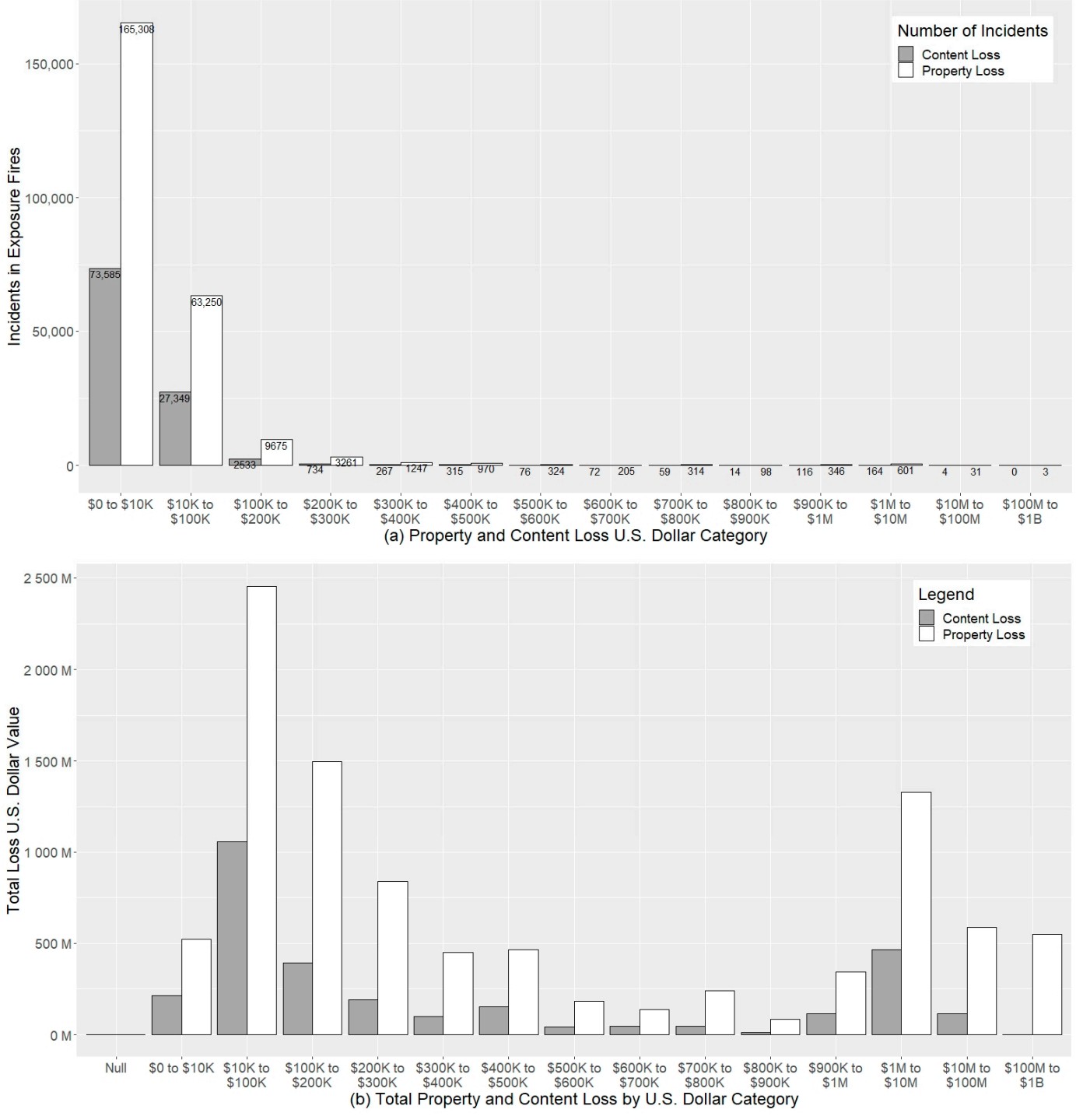

**Figure 5.** The number of incidents by property and content loss categories and the total property and content loss by the U.S. dollar category for exposure fires reported to the NFIRS between 2002 and 2020. (**a**) Number of incidents. (**b**) U.S. dollar value category (e.g., USD 0 to 10 K).

One of these incidents is a wildland fire exposure from the 2007 California Witch Fire (USD 237,401,549), possibly meant to represent all damage to wildland resources. Another incident is from a building fire in New Orleans, Louisiana, with two exposures (restaurant and food store), with one incident having a property loss of USD 200,000,000. We could not find any information on this fire, and there is no discernible damage when comparing pre-fire and post-fire Google Earth™ imagery, indicating that this might be a data entry error.

The third incident was from an exposure fire in Waltham, Massachusetts, with the source fire having a property loss of USD 110,000,000. The property loss value reported might be representative of the actual property loss. There was a five-alarm fire in this location at an under-construction apartment complex.

There are 31 incidents with property loss values between USD 10,000,000 and USD 100,000,000, totaling USD 586,307,361. These incidents might represent data entry errors or actual property loss values. For example, there are 14 incidents from a fire in Greenfield, CA, at cannabis greenhouses, each having a property loss value of USD 12,000,000, which could represent the actual dollar loss as the buildings are damaged in Google Earth™. There are fewer extreme outliers for the content loss values. However, content (164) and property (601 incidents) losses have significant spikes for incidents with dollar losses between USD 1,000,000 and USD 10,000,000 (Figure 5). We did not check these incidents to ensure they represent reasonable estimates of dollar losses for the incidents.

The percent of property loss for the 69,130 incidents where the NFIRS has this information documented is 81% for incidents with exp_no equal to zero with a median of 100% (i.e., the first incident). For the 113,385 incidents where the NFIRS documented percent property loss with an exp_no greater than 0 (i.e., incidents resulting from the first incident), the mean percent of property loss is 51%, and the median is 38%. The percent of content loss for the 44,657 incidents where the NFIRS has this information documented is 85%, with a median of 100% for incidents with an exp_no equal to zero (i.e., the first incident). For the 38,009 incidents where the NFIRS documented percent property loss with an exp_no greater than 0, the mean percent of property loss is 71%, and the median is 100%.

Single WUI fires can contribute significantly to the reported property and content loss values. For example, the 2016 Tennessee Chimney Tops 2 exposure fires with more than ten incidents reported to the NFIRS (which multiple departments reported depending on the area affected) have a total property loss value of USD 285,087,029. Nonetheless, the NFIRS-reported property and content loss values are still significant when considering only those incidents with less than or equal to ten exposures and reported property loss or content loss values between USD 10,000 and USD 1,000,000 (property loss equals USD 5,647,121,172 and content loss equals USD 1,777,345,793).

These property and content losses are not national estimates but only NFIRS-reported losses. The NFIRS data do not represent a statistically valid sample of exposure fires in the U.S., contain a small number of exposure fires compared to all fires in the NFIRS, and underreport WUI fires, as detailed below. Therefore, determining the validity of using scaling factors to derive national estimates [28] of damages, the number of national exposure fires, and other statistics requires further study.

*3.4. NFIRS Exposure Fires by SILVIS WUI Type, Actions Taken, and Heat Source*

We also portray the NFIRS exposure fires by WUI [23] classification (Figure 6) for the 219,571 incidents we could geolocate within the conterminous U.S. Some of these incidents represent features like vehicles that were geolocated by the address of the primary structure, and the location might not be precise. Twenty-seven percent of the exposure fires occur in WUI areas. The highest percentage of WUI exposure fires is in the medium-density interface, followed by low-density intermix and high-density interface WUI areas (Figure 6). A small number of exposure fires occurred in built areas close to water. The relatively coarse scale of the WUI dataset resulted in these built areas being classified as occurring in water.

Regarding defensive actions reported at the exposure fires examined, extinguishment was the most frequently reported first action, occurring in 76% (264,734 of 348,089) of the incidents (Table 1). First actions categorized as fire control and extinguishment (extinguish; fire, other; salvage and overhaul; contain wildland fire; establish fire lines; control wildland fire; confine wildland fire; manage prescribed fire) occur in 84% of the exposure fire incidents (Table 1). The NFIRS reports 86% of the incidents with a fire control and extinguishment action when considering the first, second, and third actions taken (act_tak1, act_tak2, act_tak3).

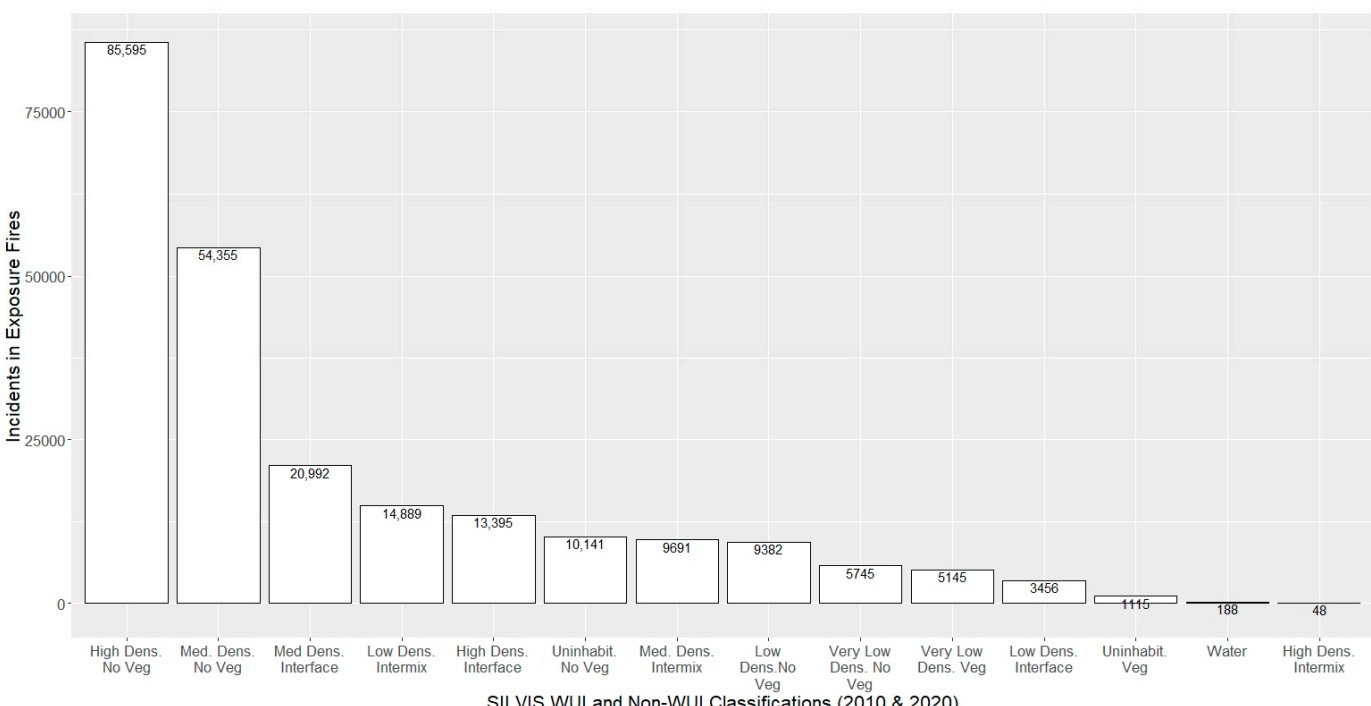

**Figure 6.** The distribution of NFIRS exposure fire incidents (2002 to 2020) by SILVIS [23] WUI classification. This total differs from the 234,137 exposure fire incidents geolocated because we do not consider Puerto Rico, Hawaii, or Alaska.

**Table 1.** Defensive action taken (act_tak1) at NFIRS exposure fires between 2002 and 2020.

| Defensive Action Taken | Number of Incidents | Defensive Action Taken1 | Number of Incidents |
|---|---|---|---|
| Extinguish | 264,734 | Secure property | 105 |
| Investigate | 22,148 | Refer to proper authority | 95 |
| Fire, other | 17,084 | Undetermined (conversion only) | 90 |
| Incident command | 13,883 | Assess severe weather or natural disaster damage | 87 |
| Salvage and overhaul | 7034 | Provide first aid and check for injuries | 86 |
| Action Taken, Other | 4970 | Shut down system | 72 |
| Investigate fire out on arrival | 2945 | Canceled en route | 53 |
| Ventilate | 2413 | Emergency medical services, other | 47 |
| Contain fire (wildland) | 1483 | Provide equipment | 47 |
| Investigation and enforcement, other | 1306 | Provide information to public | 45 |
| Establish fire lines (wildfire) | 1183 | Undetermined | 44 |
| Search | 1021 | Enforce code | 42 |
| Forcible entry | 869 | Hazardous materials control | 29 |
| Remove hazard | 784 | Manage prescribed fire (wildland) | 21 |
| Establish safe area | 622 | Fill-in, standby, other | 19 |
| Control fire (wildland) | 575 | Restore fire alarm system | 19 |
| Provide manpower | 442 | Hazmat detection | 17 |
| Rescues and hazardous conditions | 440 | Control traffic | 16 |
| Evacuate area | 410 | Recover body | 15 |

**Table 1.** *Cont.*

| Defensive Action Taken | Number of Incidents | Defensive Action Taken1 | Number of Incidents |
| --- | --- | --- | --- |
| Standby | 325 | Systems and services, other | 15 |
| Rescue, remove from harm | 274 | Transport person | 14 |
| Confine fire (wildland) | 273 | Extricate, disentangle | 13 |
| Identify hazardous materials | 256 | Restore sprinkler or fire protection system | 11 |
| Provide apparatus | 235 | Fill-in or move-up | 10 |
| Provide water | 231 | Hazardous materials spill control and confinement | 10 |
| Search and rescue, other | 183 | Control crowd | 9 |
| Notify other agencies. | 156 | Determine if materials are non-hazardous | 4 |
| Assistance, other | 147 | Provide light or electrical power | 3 |
| Remove water | 140 | Remove hazardous materials | 3 |
| Provide advanced life support (ALS) | 133 | Assist animal | 2 |
| Hazardous condition, other | 117 | Assist physically disabled | 2 |
| Operate apparatus or vehicle | 114 | Restore municipal services | 2 |
| Provide basic life support (BLS) | 111 | Provide air supply | 1 |

We present the distribution of heat source values for incidents in the FireIncident table, which has a corresponding incident in the BasicIncident for source fires (exp_no equal to 0) in Table 2 and for the resultant exposure fires (exp_no greater than 0) in Table 3. Not all records in the BasicIncident table contained corresponding records in the FireIncident table (i.e., only 327,900 of 348,089). Also, 108,684 or 33% of FireIncident records corresponding to exposure incidents had unknown values for the heat source.

**Table 2.** Distribution of heat source values for incidents with exp_no equal to 0 in the FireIncident table with a matching record in the filtered (118,089 of 348,089 as shown in) BasicIncident table.

| Heat Source | Number of Incidents | Heat Source | Number of Incidents |
| --- | --- | --- | --- |
| Unknown | 69,861 | Flame/torch used for lighting | 634 |
| Arcing | 7271 | Multiple heat sources including multiple ignitions | 614 |
| Radiated, conducted heat from operating equipment | 4999 | Heat from undetermined smoking material | 583 |
| Heat from powered equipment, other | 4757 | Chemical reaction | 520 |
| Hot ember or ash | 4364 | Backfire from internal combustion engine | 442 |
| Heat from other open flame or smoking materials | 3584 | Molten, hot material | 413 |
| Spark, ember or flame from operating equipment | 3295 | Radiated heat from another fire | 388 |
| Cigarette | 2464 | Conducted heat from another fire | 178 |
| Hot or smoldering object, other | 2440 | Chemical, natural heat source, other | 155 |
| Cigarette lighter | 2159 | Explosive, fireworks, other | 108 |
| Match | 2114 | Other static discharge | 100 |

**Table 2.** *Cont.*

| Heat Source | Number of Incidents | Heat Source | Number of Incidents |
|---|---|---|---|
| Heat from direct flame, convection currents | 1131 | Pipe or cigar | 60 |
| Heat spread from another fire, other | 914 | Sunlight | 49 |
| Incendiary device | 790 | Warning or road flare; fusee | 19 |
| Candle | 785 | Munitions | 5 |
| Heat, spark from friction | 783 | Blasting agent | 3 |
| Fireworks | 744 | Model and amateur rockets | 2 |
| Flying brand, ember, spark | 716 | Heat spread from another fire | 1 |
| Lightning | 643 | Other heat sources | 1 |

**Table 3.** Distribution of heat source values for incidents with exp_no greater than 0 in the FireIncident table with a matching record in the filtered (209,811 of 348,089 incidents) BasicIncident table.

| Heat Source | Number of Incidents | Heat Source | Number of Incidents |
|---|---|---|---|
| Radiated heat from another fire | 57,977 | Fireworks | 327 |
| Heat from direct flame, convection currents | 47,347 | Incendiary device | 315 |
| Unknown | 38,823 | Molten, hot material | 296 |
| Heat spread from another fire, other | 28,804 | Candle | 263 |
| Heat from other open flame or smoking materials | 6454 | Flame/torch used for lighting | 253 |
| Conducted heat from another fire | 6214 | Heat from undetermined smoking material | 247 |
| Flying brand, ember, spark | 4604 | Chemical reaction | 173 |
| Hot ember or ash | 3952 | Backfire from internal combustion engine | 122 |
| Arcing | 2216 | Explosive, fireworks, other | 71 |
| Hot or smoldering object, other | 2175 | Chemical, natural heat source, other | 57 |
| Radiated, conducted heat from operating equipment | 1965 | Other static discharge | 46 |
| Heat from powered equipment, other | 1645 | Sunlight | 23 |
| Spark, ember or flame from operating equipment | 1224 | Pipe or cigar | 16 |
| Multiple heat sources including multiple ignitions | 1167 | Warning or road flare; fusee | 14 |
| Cigarette | 846 | Model and amateur rockets | 13 |
| Match | 701 | Munitions | 10 |
| Cigarette lighter | 658 | Blasting agent | 6 |
| Heat, spark from friction | 423 | Heat spread from another fire | 1 |
| Lightning | 362 | Other heat sources | 1 |

Values for the heat_sour attribute when the exp_no is greater than 0 should correspond to a value between 80 and 84, representing heat spread from another fire, direct flame, radiated heat, embers, or conducted heat from another fire. However, 64,865 incidents (31%) with an exp_no greater than 0 have heat_sour values not between 80 and 84 (Table 3). Some of these values are logical heat sources for exposure fires, such as heat from other

open flame or smoking materials and hot embers or ash. Slightly over two percent (4604 of 209,811) of known heat source values for incidents where exp_no was greater than 0 are from flying brands, embers, or sparks.

### 3.5. Examination of NFIRS Exposure Fires with Significant Incidents

Next, we examine all exposure fires with more than 20 incidents and more than 10 incidents for single/multifamily residence exposure fires (final filter in Figure 2). We classified 45% (91 of 201) of these NFIRS-reported exposure fires as WUI fires (Figure 7). Following WUI fires, the most significant exposure fires examined (regarding the number of incidents) were suburban/urban, apartment, suburban, and urban fires (Figure 7). We present the full table of these exposure fires in Appendix B.

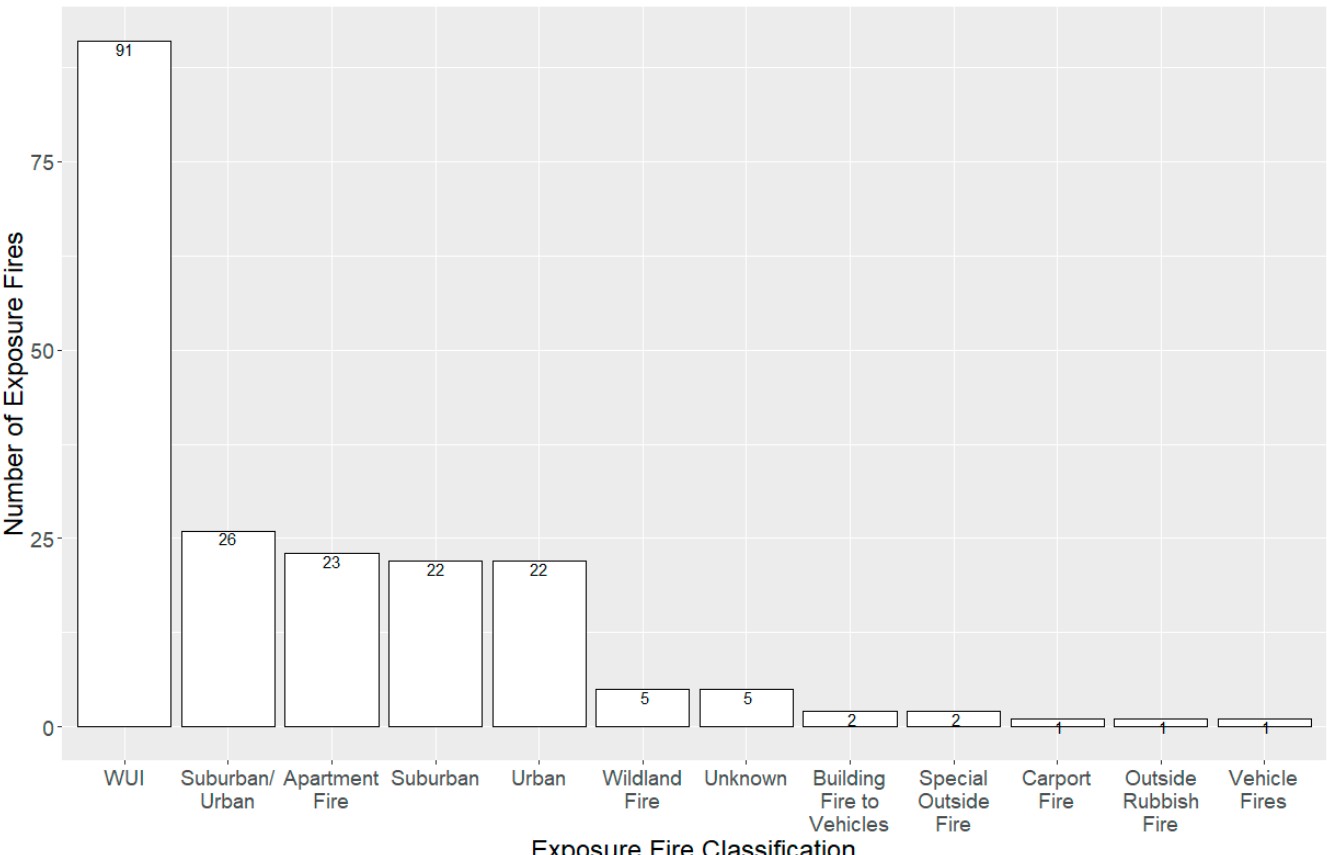

**Figure 7.** Distribution of exposure fires with greater than 20 incidents or more than ten incidents containing single/multifamily residences.

Five of these significant exposure fires contained all wildland fire exposures. Five fires could not be classified. Two fires started in structures and spread to vehicles; two were special outside fires. One fire was a carport fire in an apartment building involving multiple vehicles, one was an outside rubbish fire, and one was a fire involving multiple vehicles only (Figure 7).

For the WUI fires, we compared the affected buildings listed in the SIT-209 fire database or other sources (e.g., CAL FIRE Incident Reports or media reports for fires we could name) to the NFIRS (Table 4). Typically, these NFIRS exposure fires affect fewer buildings. In some cases, the NFIRS exposure fires underreported the number of buildings affected. However, other sources can also underreport the affected buildings compared to the NFIRS.

**Table 4.** Comparison of NFIRS-reported building fires (2002 to 2020) in WUI exposure fires against reported building fires from other sources.

| Fire Name, State, and Date (mm/yyyy) | NFIRS Building Affected | Buildings Affected Other Sources | Difference |
| --- | --- | --- | --- |
| TN Chimney Tops 2 Fire 11/2016 | 1873 | 2460 | −587 |
| CA Witch Fire 10/2007 | 473 | 1736 | −1263 |
| CA Valley Fire 9/2015 | 218 | 2051 | −1833 |
| CA Harris Fire 10/2007 | 320 | 563 | −243 |
| CA Freeway Complex Fire 11/2008 | 176 | 361 | −185 |
| CA Clayton Fire 8/2016 | 297 | 328 | −31 |
| CA Humboldt Fire 6/2008 | 117 | 261 | −144 |
| CA Rice Fire 10/2007 | 208 | 248 | −40 |
| CA Clover Fire 9/2013 | 59 | 211 | −152 |
| CA Boles Fire 9/2014 | 242 | 172 | 70 |
| AR Chaffee Fire 1/2008 | 34 | 150 | −116 |
| AR Chaffee Fire 1/2008 | 121 | 150 | −29 |
| CA Poomacha Fire 10/2007 | 15 | 217 | −202 |
| MN Ham Lake 5/2007 | 163 | 133 | 30 |
| SC Windsor Greens Fire 3/2013 | 26 | 26 | 0 |
| CA BTU Lightning Comp. Fire 8/2008 | 119 | 117 | 2 |
| CA Summit Fire 5/2008 | 86 | 99 | −13 |
| AK McKinley Fire 8/2019 | 127 | 275 | −148 |
| CA Courtney Fire 9/2014 | 51 | 56 | −5 |
| MN Green Valley Fire 5/2013 | 39 | 55 | −16 |
| AK Sockeye Fire 6/2015 | 170 | 99 | 72 |
| TN Black Bear Cub Fire 3/2013 | 73 | 73 | 0 |
| TX Ringgold Texas Fire 1/2006 | 49 | 40 | 9 |
| CA Ophir Fire 6/2008 | 25 | 49 | −24 |
| WI Germann Road Fire 5/2013 | 66 | 47 | 19 |
| CO Coal Seam Fire 6/2002 | 17 | 44 | −27 |
| CA Cocos Fire 5/2014 | 51 | 40 | 11 |
| CA Round Fire 2/2015 | 41 | 48 | −7 |
| CA Telegraph Fire 7/2008 | 131 | 130 | 1 |
| OK Harrah Fire 3/2011 | 22 | 39 | −17 |
| NV Caughlin Fire 11/2011 | 40 | 29 | 11 |
| NE Valentine Fire 7/2006 | 22 | 42 | −7 |
| TX Willow Creek Fire 2/2011 | 39 | 29 | 10 |
| TX Tanglewood Fire 2/2011 | 69 | 48 | 21 |
| CA Stagecoach Fire 8/2020 | 18 | 60 | −42 |
| CA Trabing Fire 6/2008 | 93 | 20 | 73 |

**Table 4.** *Cont.*

| Fire Name, State, and Date (mm/yyyy) | NFIRS Building Affected | Buildings Affected Other Sources | Difference |
|---|---|---|---|
| TX Pitt Road Fire 5/2011 | 10 | 20 | −6 |
| MT Roaring Lion Fire 7/2016 | 18 | 16 | 2 |
| NM Quail Ridge Fire 2/2011 | 17 | 15 | 2 |
| CA Vail Fire 9/2009 | 5 | 15 | −10 |
| CA Lockheed Fire 8/2009 | 12 | 14 | −2 |
| GA Sweat Farm Again Fire 6/2011 | 17 | 14 | 4 |
| ID Sweetwater Fire 8/2008 | 26 | 21 | 5 |
| CA Washoe Fire 8/2007 | 8 | 6 | 2 |
| WA Boffer Fire 8/2018 | 11 | 9 | 2 |
| FL Lincoln and 6th St. Fire 5/2013 | 14 | 3 | 11 |

There are 21 fires where the SIT-209 database or other sources underreports the number of buildings affected compared to the NFIRS-reported buildings affected, representing 605 buildings (Table 4). There were 26 fires where the NFIRS underreported the buildings affected compared to the SIT-209 database or other sources, representing 4124 buildings (Table 4). Additionally, there were 37 WUI fires contained in the NFIRS where we could not find any information about them from other sources (Appendix B). Additional information on the number of buildings affected by these fires might be in some statewide databases not examined here (e.g., the California Incident Data and Statistics Program).

Furthermore, the NFIRS WUI exposure fires do not contain information on many of the largest WUI fires documented in the SIT-209 database between 2002 and 2020, including the 2018 California Camp, 2017 California Tubbs, 2003 California Cedar, 2015 California Valley, 2011 Texas Bastrop, 2012 Colorado Waldo Canyon, and 2014 Colorado Black Butte Fires, to name a few. However, the NFIRS-reported exposure fires also contain WUI fires not reported in the SIT-209 database or other sources examined. These exposure fires not contained in other sources, such as the 2008 Oregon Trail Fire in Boise, ID (Appendix B), sometimes represent WUI fires not documented elsewhere other than NFIRS or media reports.

Underreporting of structures affected can occur for many reasons. For example, the underreporting of the 2007 California Witch Fire is partly due to some fire departments not reporting to the NFIRS. For example, Poway and San Diego do not have any exposure fires reported for the Witch Fire (Figure 8). However, surrounding cities contain incidents corresponding to damage and destruction from the Witch Fire (Figure 8).

We further examine the extent of underreporting in NFIRS exposure fires at the 2011 Texas Tanglewood Fire. This fire was the subject of a detailed assessment of damaged features from a post-fire evaluation [25,26]. Overall, the detailed post-fire assessment reported significantly more features as damaged and destroyed than in the NFIRS and what was reported in the SIT-209 database (Table 5).

The NFIRS contained two exposure fires with incidents from the 2011 Tanglewood Fire, one of which the Amarillo Fire Department reported as mutual aid (aid equals 4). Mutual aid fires are typically excluded from NFIRS analysis to avoid double counting. However, in this case, the incidents coded as being from mutual aid were not double-counted, representing unique components of the exposure fire. The NFIRS significantly underreported the features affected by the 2011 Tanglewood Fire (Table 5).

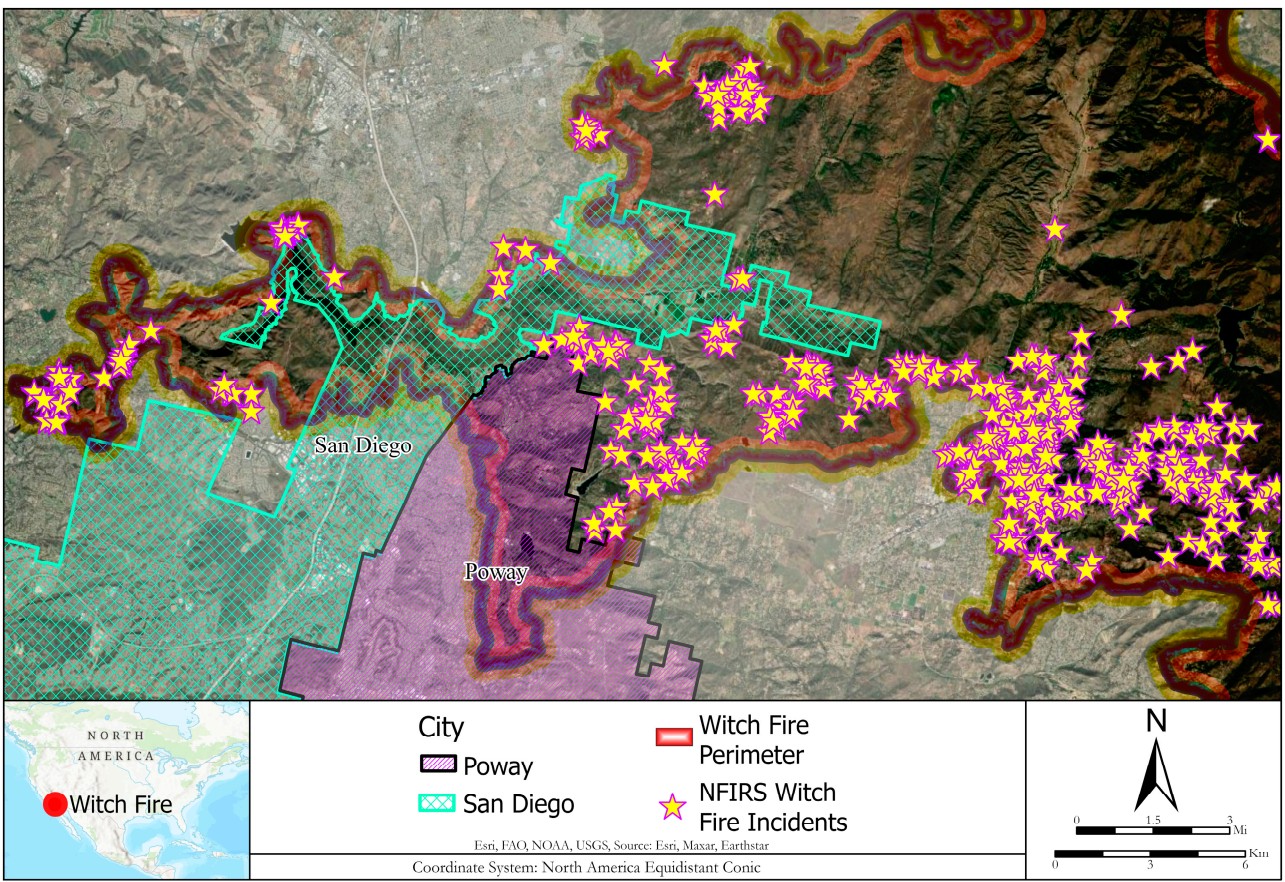

**Figure 8.** The differences in NFIRS-reported exposure fires based on jurisdictional boundaries (cities) at the 2007 California Witch Fire [29] contributed to the underreporting of fire-affected features.

**Table 5.** Comparison of damaged or destroyed features reported by NFIRS and a detailed post-fire assessment [25,26] of the 2011 Texas Tanglewood Fire.

| Fire | Damage Assessment Method | Count of Damaged or Destroyed |
|---|---|---|
| | NFIRS | 2 |
| Single-Family Residences | Post-Fire [25] | 48 |
| | Difference | 46 More Post-Fire |
| | NFIRS | 69 |
| Unclassified Buildings | Post-Fire [25] | 0 |
| | Difference | 69 More NFIRS |
| | NFIRS | 4 |
| Outbuildings | Post-Fire [25] | 78 |
| | Difference | 74 More Post-Fire |
| | NFIRS | 2 |
| Vehicles | Post-Fire [25] | 31 |
| | Difference | 29 More Post-Fire |
| | NFIRS | 0 |
| Fences, Retaining Walls, and Other Linear Features | Post-Fire [25] | 108 |
| | Difference | 108 More Post-Fire |

**Table 5.** *Cont.*

| Fire | Damage Assessment Method | Count of Damaged or Destroyed |
|---|---|---|
| | NFIRS | 0 |
| Other Features | Post-Fire [25] | 241 |
| | Difference | 241 More Post-Fire |
| Total Difference | | 429 More Post-Fire |

However, these underreported features represented secondary features like fences, retaining walls, wood piles, secondary structures, or other relatively small features. It would be onerous for fire departments to capture all these fire-affected features in even moderately sized WUI incidents. For example, multiple field crews and one office team spent several weeks capturing the full extent of fire-affected features at the 2011 Texas Tanglewood Fire [25,26]. Nevertheless, features such as wood piles, fences, and others were shown to be fire hazards, contributing to fire spread between structures and from the wildlands to structures [26].

*3.6. NFIRS Structure Separation Distance*

We present the results of examining SSD between damaged-to-damaged and damaged-to-not-damaged structures (Figure 1) within 200 m of damaged structures in Figure 9. There were 190 of the 17,774 (7%) damaged structures, with an SSD for the closest damaged structure greater than 200 m and not included in this examination. There were 935,399 not damaged structure footprints within 200 m of the damaged structures.

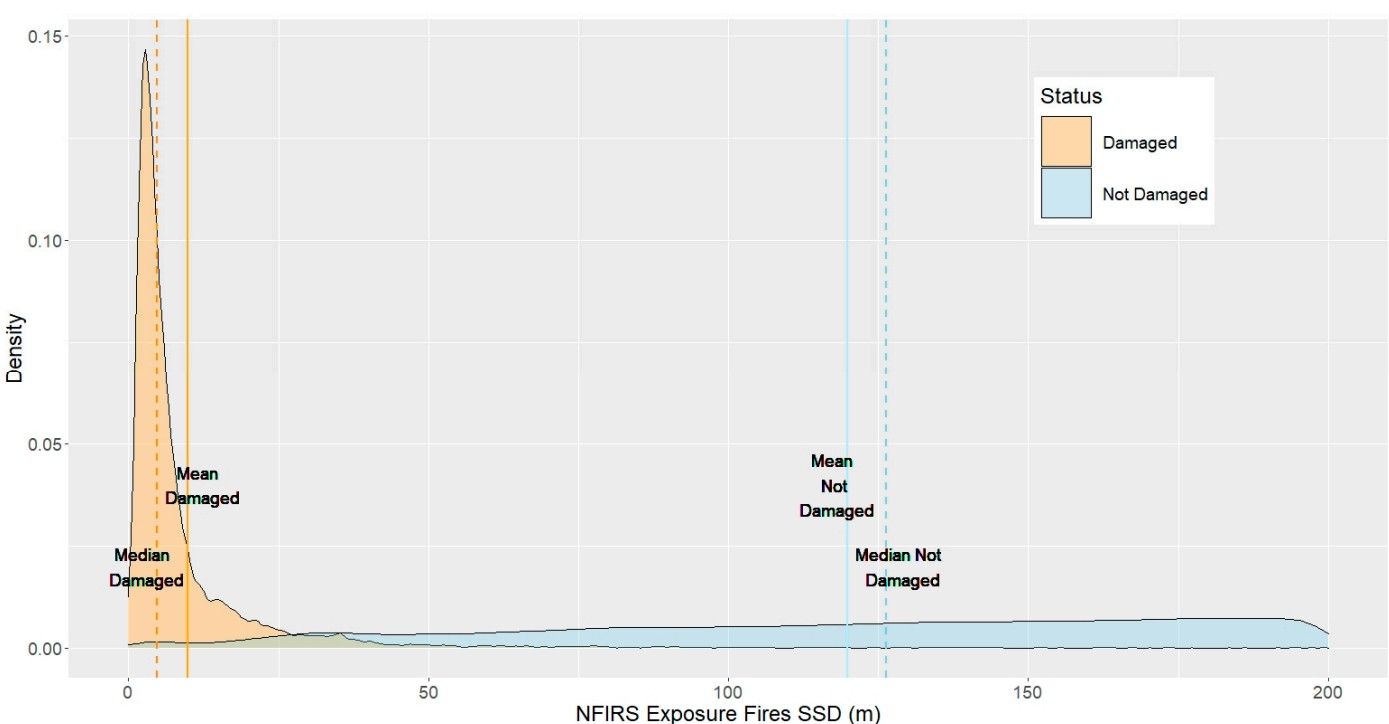

**Figure 9.** The density plots of SSD between damaged-to-damaged buildings versus damaged-to-not-damaged buildings as reported in assessed NFIRS exposure fires between 2002 and 2020.

The distribution of damaged-to-damaged SSD for those structures with an SSD less than 200 m is positively skewed (Figure 9). Considering damaged structures with an SSD less than or equal to 200 m, the probability of finding a damaged structure near another damaged structure peaks when the SSD is about 3 m. Alternatively, the distribution of damaged-to-not-damaged structures with an SSD to the nearest damaged structure less

than or equal to 200 m is relatively flat, generally increasing to about 190 m (Figure 9). Likely, the drop-off at 190 m is an edge effect, with the general increase representative of more structures present the more significant the distance.

We confirm the difference between the SSD distributions of the two populations by performing a two-sample Kolmogorov–Smirnov test. The result of this test is a $p$-value $< 2.2 \times 10^{-16}$, indicating that we have sufficient evidence to say the two samples do not come from the same distribution. The damaged sample has a mean SSD of 6.5 m and a median of 4.4 m, and the not-damaged sample has a mean of 18.1 m and a median of 20.3 m.

The one-hundred and ninety damaged structures with an SSD to the closest damaged structure greater than 200 m sometimes occurred when the damage was not necessarily a result of structure-to-structure fire spread. For example, this situation might occur in WUI fires where the fire department only recorded the wildland fire incident once. However, the wildland fire might have ignited more than one structure in these cases, greater than 200 m apart.

Also, there might have been other undocumented incidents (e.g., fences) between the damaged structures that contributed to the fire spread. In some cases, it is unclear if the incidents in these exposure fires with large SSDs were related and perhaps only occurred at similar times, being incorrectly recorded as occurring in the same incident. Furthermore, some incidents could have had far-field ember spread. For example, a large fire in Overland Park, Kansas, in March of 2017 showed evidence of embers igniting structures over significant distances (Figure 10).

The distances shown in Figure 10 represent the minimum distance between incidents in the exposure fire. The embers' specific source is unknown, though it was from structures. Both southern incidents shown in Figure 10 were residential structures with wood shake roofs, as seen in Google Streetview™. Post-fire imagery in Google Earth™ shows a change in the roof type. An overhead fire video [30] shows the second most southern structure (Figure 10) ignited after most other northern structures were burning. The ignition of this structure was in the roof area. Finally, first responders documented that wood shake roofs and dry conditions made containment difficult [31].

In this exposure fire (Figure 10), four incidents are documented by a fire department as aid given. Two of these incidents occurred on the same structure and two on separate structures not documented by the fire department having jurisdiction. Excluding those incidents that are double counted, 15 of the 33 incidents (46%), the NFIRS reports a heat_sour value of flying brand, ember, or spark.

Other examples of potential far-field ember spread are present in NFIRS exposure fires. These examples include a fire in Richmond, VA, on 26 March 2004, involving numerous features and ember spread up to 100 m; a WUI fire in Oklahoma City, OK on 9 April 2009, involving six structures with ember spread up to 140 m; a fire in Yuma, AZ on 29 January 2018, involving eight structures with ember spread beyond 300 m; and many others. The FireIncident table documents 669 of 17,774 incidents (3.8%) as being ignited by hot ember, ash, flying brands, embers, or spark, with an additional 145 incidents (0.8%) ignited potentially from embers (i.e., spark, ember, or flame from operating equipment; or heat, spark from friction). However, understanding the details of the ember spread is challenging without other sources such as those cited above [30,31] and identified in Figure 10.

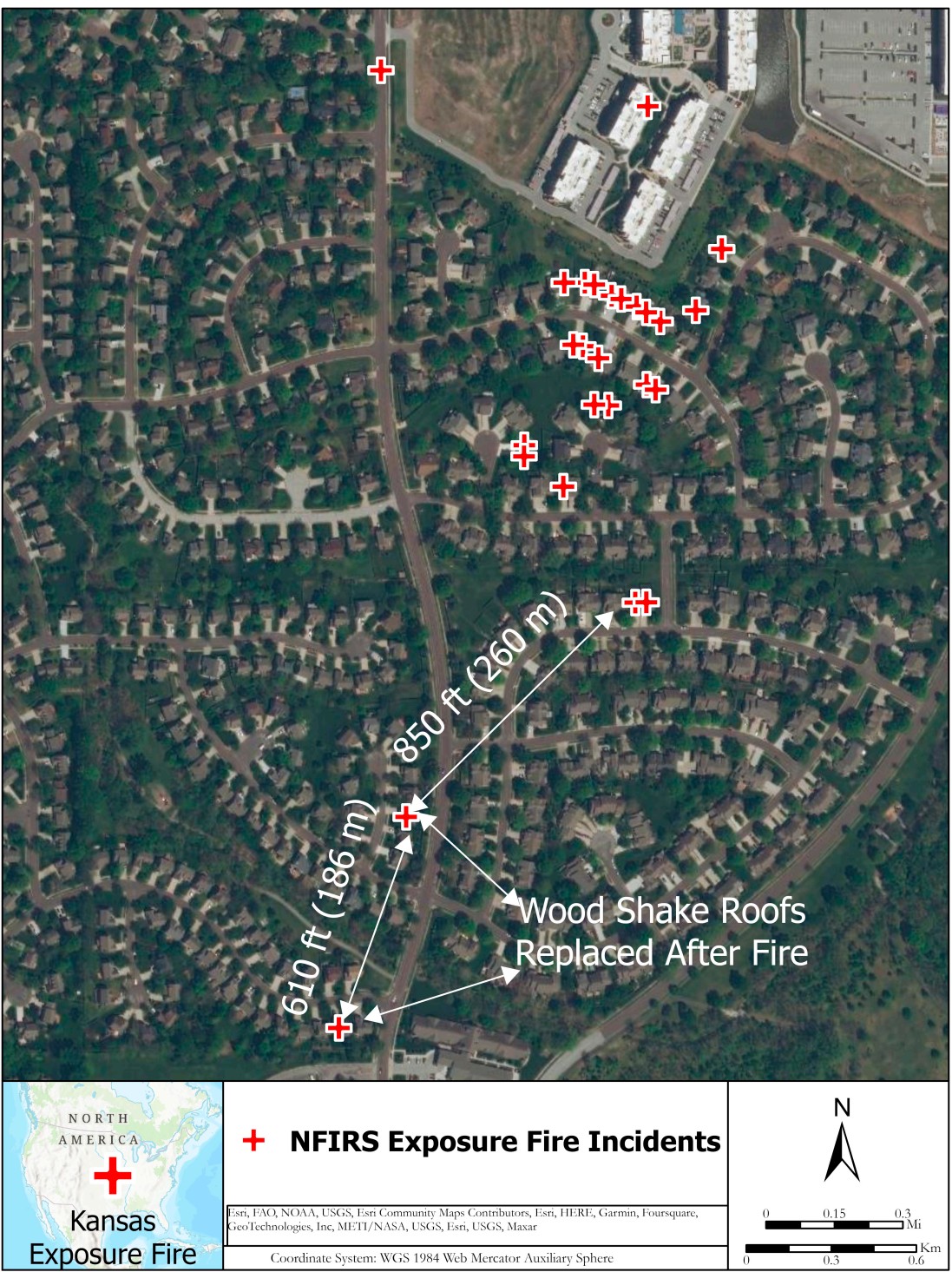

**Figure 10.** An NFIRS-documented exposure fire in Overland Park, Kansas on 20 March 2017. The exposure fire started in the apartment building at the north end of the map.

### 3.7. NFIRS Ignition Pathways

We present the results of our examination of ignition pathways for two incident exposure fires in Figure 11. When the first incident type is a structure, the most significant number of incidents for the second one is also a structure. This pattern is the same for vehicles, outside, and other fires.

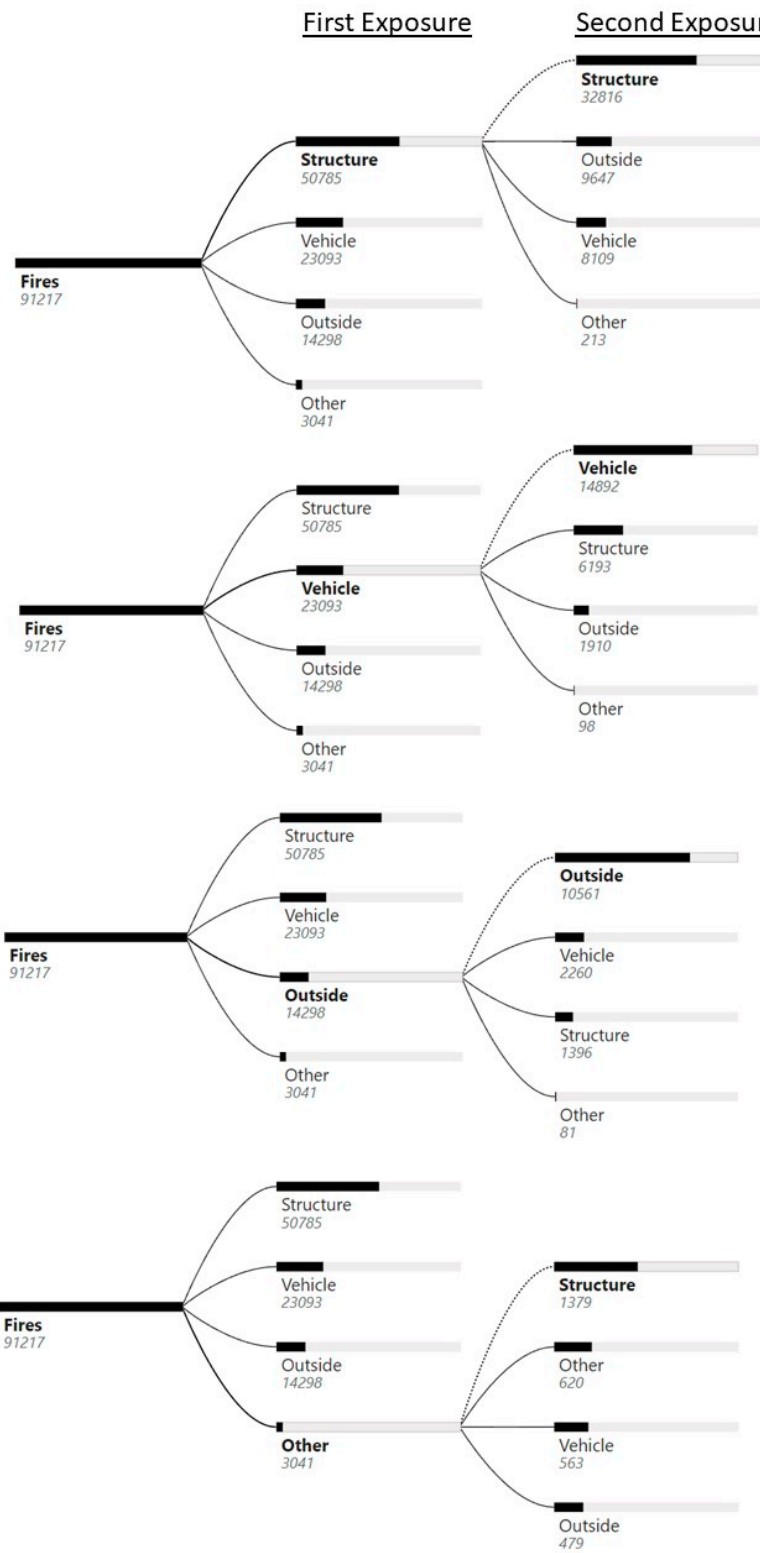

**Figure 11.** The general incident types for exposure fires with two incidents.

## 4. Discussion

We presented an extensive assessment of exposure fires from the NFIRS database, covering 19 years of reporting. Even when excluding large WUI fires not in the NFIRS, exposure fires represent a significant fire problem based on NFIRS-reported dollar losses over the period examined (conservatively, USD 5,647,121,172 in reported property losses and USD 1,777,345,793 in content losses). NFIRS-reported exposure fires primarily involve

buildings, vehicles, and natural vegetative fires in residential, outside, and storage areas (Figure 4). However, other features in different property types can be involved.

Some losses from NFIRS exposure fires (Appendix B), particularly for exposure fires with limited incidents (e.g., less than ten), represent losses from WUI fires not reported in databases such as the SIT-209. The documentation of these smaller WUI fires can enable comparisons to WUI fires where destruction to human-made features is significant, thereby aiding our understanding of conditions that lead to substantial structure destruction, including encroachment of vegetative fires into urban or suburban areas not designated as WUI and structure-to-structure fire spread once a wildland fire enters a community. The NFIRS also reports on damage to features other than buildings, which other national databases in the U.S. do not, enabling the study of the effect of these features on exposure fires. A significant percentage (74%) of the exposure fires examined also occur outside areas mapped as WUI (Figure 6).

However, at a fine scale, these non-WUI areas can look similar to areas mapped as WUI, with arbitrary lines that fire easily crosses, sometimes delineating the two environments (e.g., Figure 12). The exposure fires in Figures 1 and 10 do not occur in WUI areas. Nonetheless, the environments are similar to areas in the built environment destroyed by WUI fires, such as those at the 2012 Colorado Waldo Canyon Fire [16], the 2007 California Witch Fire [29], the 2011 Texas Tanglewood Fire [26], the Coffee Park Neighborhood Fire (Figure 12), and many others.

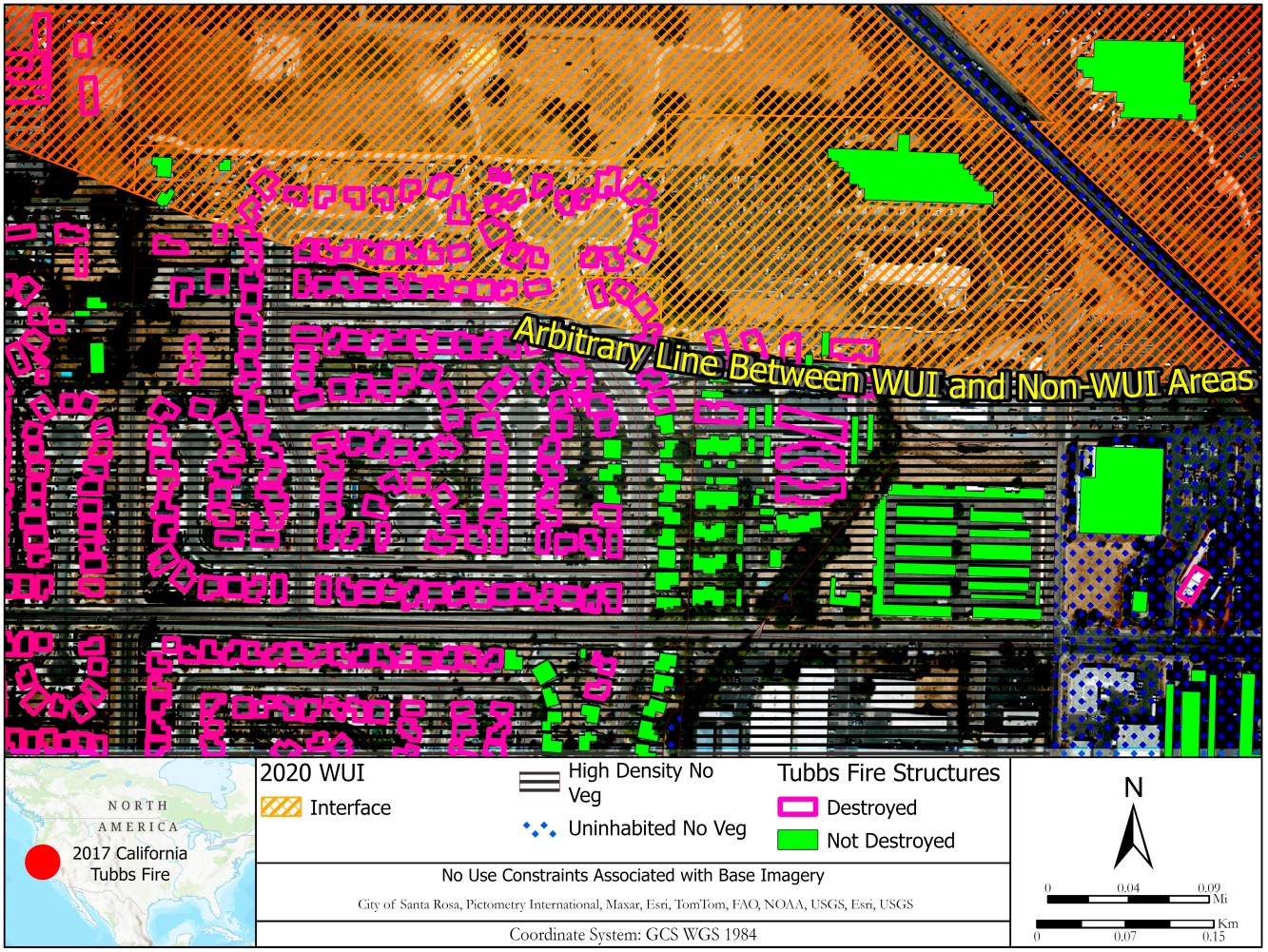

**Figure 12.** WUI mapping classifications [23] show an arbitrary line between WUI and non-WUI areas for the Coffey Park neighborhood affected by the 2017 Tubbs Fire (data from [15]), which was not documented in the NFIRS. Uninhabited areas contained commercial buildings.

Many NFIRS exposure fires occur in high-density areas with limited wildland vegetation (Figure 6), highlighting the potential for large structure conflagrations in these areas, particularly with changing climate conditions, increased weather severity (e.g., wind and drought), and diminishing resources (e.g., gasoline and water). For example, as water becomes scarce in areas such as the U.S. desert southwest, suppression capabilities might be hindered, requiring increased vigilance regarding reducing initial ignitions in regions with historically high numbers of exposure fires.

Also, tracking exposure fires can aid in evaluating "Let it Burn" policies in urban areas with many abandoned structures. For example, the Detroit, MI area contains one of the highest densities of geolocated NFIRS exposure fires examined. This high density could partly be due to abandoned structures in this area and "Let it Burn" policies implemented to focus scarce firefighter resources [32]. However, complete or incorrect reporting issues hinder statistical analysis of NFIRS exposure fires.

For example, a large number (108,684 or 33%) of the NFIRS-reported heat sources for exposure fires were missing. Also, NFIRS procedures list five acceptable heat source codes (80–84) representing heat spread from another fire for incidents in exposure fires after the source fire (i.e., exp_no greater than 0). These heat source codes capture direct flame and convection currents, radiated heat, embers, conducted heat, and other heat spread. However, not all documented heat sources were of these types, though some were similar, such as a spark, ember, or flame from operating equipment.

Radiated heat from another fire was the most numerous heat source for incidents with exp_no greater than zero. The next most numerous heat source was direct flame or convection currents, followed by unknown, heat spread from another fire (other), heat from open flame or smoking material, conducted heat from another fire, and flying brand, ember, spark, and hot ember or ash. Regarding the population of incidents (17,774 as shown in Figure 2) used to assess SSD, a small percentage (2.2%) of the known heat_sour values were recorded as being from ember spread from another fire.

Some attributes, however, are reported more thoroughly than heat sources, such as defensive actions. Extinguishment occurs at most NFIRS exposure fire incidents (86%). The more extensive reporting of actions taken than heat sources could be because first responders were aware of their actions. However, the heat source that caused the ignition of other features might have been more difficult to discern and potentially confusing to document. Difficulties documenting heat sources at exposure fires highlight our lack of understanding of ignition mechanisms and the source of heat fluxes at exposure fires, even when they contain a limited number of features.

We also expanded upon the previously documented [19] underreporting of NFIRS WUI fires, detailing the extent to which the large WUI fires reported in the NFIRS as exposure fires compared to estimates from other sources of damage and destruction (Table 4 and Appendix B). Our method of extracting exposure fires, in some cases, identified NFIRS WUI fires not previously reported in other studies. For example, Butry and Thomas [19] identified zero structures reported in the NFIRS from the Witch, Rice, and Poomacha Fires. In contrast, the methods employed here identified 473, 208, and 15 buildings damaged from these three fires, which still significantly underestimates the buildings damaged by these fires.

Our analysis highlights that mutual aid incidents do not always seem to double count the incidents in exposure fires. At WUI fires, the fire department responding with mutual aid is responding to incidents that the local fire department might not have responded to due to their resources responding to other incidents. Therefore, excluding mutual aid fires from NFIRS exposure fires might further exacerbate the underreporting of WUI fires in the NFIRS because the method excludes incidents that are not double-counted in some cases. Having consistent location information for incidents would help identify when double counting occurs and when it does not, helping to identify exposure fires consistently in the future.

Many (45%) of the most significant exposure fires (incidents greater than ten) reported in the NFIRS are from WUI fires. Nonetheless, the NFIRS did not capture most large WUI fires from 2002 to 2020 as exposure fires (e.g., the 2018 California Camp Fire). When the NFIRS reports large WUI fires, they are often underreported, sometimes due to differences in reporting between jurisdictions (Figure 8). In many cases, such as the 2011 Texas Tanglewood Fire, we document extensive underreporting of fire-affected features, highlighting the challenges for fire departments in documenting all the damage and destruction at moderate to large-sized exposure fires. More complete documentation of fire-affected features at the Tanglewood Fire required months (in terms of human-hours spent) of field data collection and office analysis using remote sensing data [25,26].

Evaluating SSD in NFIRS exposure fires is challenging. A significant issue curtailing the ability to calculate the separation distance between damaged features in NFIRS exposure fires was an inability to geolocate the feature. For example, the significant reduction in exposure fires (60%) and incidents (71%) resulting from the filters to identify single/multiple family residences (Figure 2) was partly a geolocation issue because the NFIRS does not provide precise geolocation information for many fire incidents (e.g., vehicles). In some cases, improper geolocation could have resulted in the geolocated point being outside the building footprint, preventing a calculation of the SSD. The NFIRS geolocation information could be correct, but the geolocation engine failed to geolocate the address in the center of the footprint. Documenting precise location information, however, would further burden fire departments.

Despite the significant data reductions, this study examined SSD between single/multi-family structures across thousands of exposure fires, identifying a significant difference, though not necessarily representative of the national exposure fire problem, in the SSD distributions between the damaged-to-damaged and not-damaged-to-damaged structures within 200 m of the damaged structures. The probability of finding a damaged structure near another damaged structure peaks when the SSD is about 3 m. However, this peak should not be used to indicate a safe SSD in exposure fires, as many factors can affect an appropriate SSD to curtail fire spread. For example, the significant difference in SSD distributions is partly due to the fire containment by first responders.

The large percentage (86%) of exposure fires where defensive actions could have altered heat fluxes coupled with the difference in the mean percent of property loss for the first exposure (81%; exp_no equals 0) and the mean of property loss for subsequent incidents (51%; exp_no greater than 0), highlights that exposure fires occur when fires in the initial incident produce more destruction. This pattern is due, in part, to a lack of containment of the initial fire. The fact that subsequent exposures have a lower percentage of property loss (51% compared to 81%) suggests more successful fire containment. Therefore, most of the not-damaged buildings within 200 m had fire spread curtailed by these defensive actions (potentially coupled with other factors not assessed here, such as wind speed, topography, fuels between the structures, and others), resulting in the increasing probability of a not-damaged building occurring the further it is from a damaged building, as shown in Figure 9.

This intuitive finding is similar to previous studies of larger WUI fires that have also found a relationship between building survival and the distance to the closest destroyed building (e.g., [11,12]). These studies assume limited or no defensive actions. However, indicators of defensive actions [15,16] exist throughout the post-fire imagery [33,34] and in the damage inspections from the California Department of Forestry and Fire (CAL FIRE).

For example, post-fire imagery of the Woolsey Fire (Figure 13) shows similar indicators of defensive actions as those seen in the destroyed structure (exp_no equals 0) in Figure 1 (i.e., a darkened appearance of some destroyed structures due to the cessation of the combustion process through extinguishment), and similar to those documented at other exposure fires [15,16]. Suppose structure fires are allowed to burn without suppression. In that case, they have a white appearance when containing mostly combustible items, indicative of white ash produced by complete combustion when a building and its contents

burn more completely, as highlighted in one of the upwind structures shown in Figure 13. Also, damaged buildings are a well-established [9,15,16,26,29] indicator of defensive actions, further validated in this study of thousands of fires.

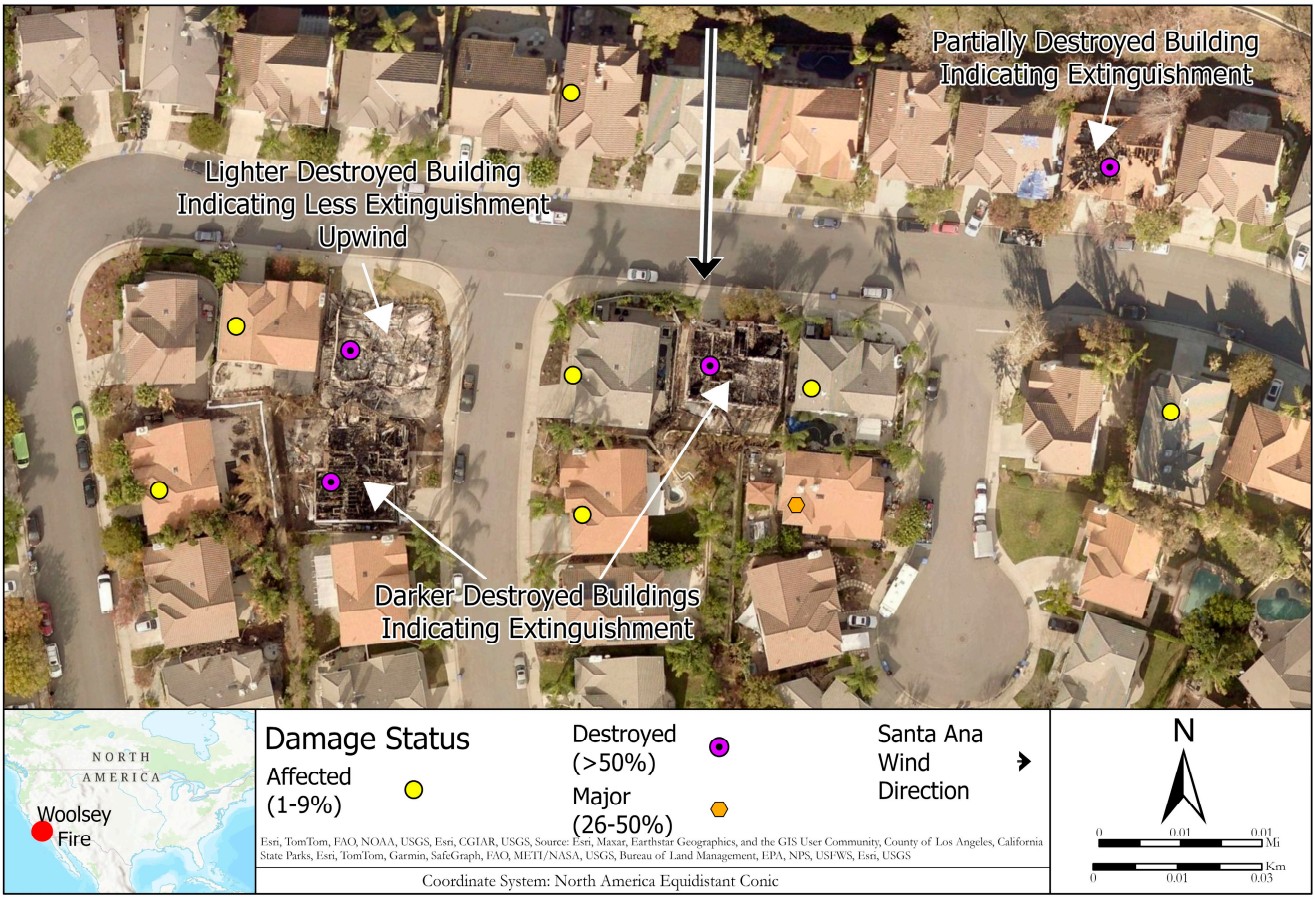

**Figure 13.** Indicators of defensive actions from Pictometry™ post-fire imagery [33] in a high-density area at the 2018 Woolsey Fire like homes with documented defensive actions in Figure 1. Partially burned buildings [35] with a darker appearance than their upwind [11] neighbors and damaged buildings, often adjacent to destroyed buildings [35], are established [15,16,26] indicators of defensive actions, further validated in this study.

Consequently, the finding that the distance to the destroyed building is the most important predictor of structure survival at the Woolsey Fire [11] is partly due to defensive actions. Without the defensive actions shown in Figure 13 and present throughout high-density areas at the Woolsey Fire and other locations, there is no evidence that fire spread between structures in these areas, particularly high-density areas (SSD less than 3 m), would have ceased. Nevertheless, as in other studies [12], this study of the Woolsey fire [11] assumed that defensive actions were limited and did not include them as a predictor variable.

Analyzing NFIRS exposure fires identifies the importance of including the effects of defensive actions, which are often assumed not to be relevant (e.g., [11,12]) when analyzing larger WUI fires. As shown here, defensive actions are prevalent in exposure fires nationwide. Large exposure fires can overwhelm defensive resources, as they did at the Woolsey Fire. Nonetheless, defensive actions can still be significant. They might be necessary to curtail fire spread in large exposure fires, as also shown in other studies [15–17,26,29].

Other factors not examined here might also contribute to preventing or reducing fire spread in exposure fires, including weather (e.g., wind), building material, and undocumented features between the structures. Consequently, obtaining a quantitative understanding of the role of low SSD from current NFIRS exposure fires would be challenging

across significant extents or even at individual fires without ancillary information (e.g., videos, images, eyewitness accounts, and measurements of heat fluxes and meteorological conditions).

Additionally, we are not sure that building damage resulted from fire spread directly between single/multifamily residences in all cases. There might have been undocumented fire incidents between structures that caused damage. For example, in some cases, we observed fences before and not after the fire in Google Earth™ Imagery between two or more single/multifamily structure incidents. The fences might have ignited, resulting in subsequent structural damage. The fire department might not have documented these more minor incidents as observed at the 2011 Texas Tanglewood Fire. This uncertainty highlights the need for experimental efforts (e.g., [36]) to understand better the role of SSD in fire spread. However, researchers can use databases like the NFIRS to help guide experiments.

Caution should be used in quantitatively interpreting results on ignition pathways as the potential underreporting of minor features between fire incidents (e.g., fences between homes) could be underestimated. Also, underreporting of large WUI fires curtails deciphering ignition pathways from NFIRS exposure fires as representative of the distributions of ignition pathways across the U.S. Nonetheless, we highlight situations (e.g., NFIRS-reported exposure fires with two incidents) where the NFIRS potentially identifies ignition pathways.

Interestingly, when the first incident in these two exposure fires is a structure, the second incident tends to be a structure, and this same pattern holds for vehicles, outside fires, and other fires. This similarity could be because features of similar types are found next to each other. This finding supports the role of smaller SSDs in fire spread, as these features must be in proximity to ignite each other. Nonetheless, NFIRS-reported exposure fires identify many possible ignition pathways, highlighting that fires do not always spread to the same feature. Also, future studies might utilize the alarm attribute, which theoretically identifies the time of the ignition to identify ignition pathways in fires with more than two incidents. However, recreating these fire timelines to capture ignition pathways precisely is challenging.

NFIRS exposure fires also provide information on ember spread at exposure fires, particularly when combined with ancillary information. For example, when integrated with active-fire overhead imagery [30] and pre-fire Google Earth™ aerial and StreetView imagery, we could confirm structure-to-structure ember spread over a significant distance (at least 260 m), also highlighting the risk of wood shake roofs in ember fire spread. Integrating these exposure fires with ancillary data, such as images and videos, can increase the understanding of factors and mechanisms affecting fire spread in exposure fires.

In many cases, first responders cannot respond to many incidents at large exposure fires because the number of ignitions overwhelms resources. Therefore, exposure fires with numerous incidents tax the NFIRS [21]. Nonetheless, if resources exist, information from exposure fires, including smaller WUI fires, could be captured using the NFIRS or NFIRS modernization efforts such as the National Emergency Response Information System (NERIS) [37], which currently aims to provide predictive analytics for WUI fires. Combined with the SIT-209 database, which portrays structure destruction from large WUI fires, the NFIRS and NERIS can add to our understanding of WUI fires by providing information on smaller fires not documented in the SIT-209 database. This approach would require a mechanism to identify incidents between different databases (e.g., NFIRS, NERIS, and SIT-209).

The NFIRS or NERIS could also be used to document select incidents from large WUI fires that are responded to by fire departments, thereby efficiently providing some data for post-fire studies that have required significant effort to obtain defensive action information (e.g., [26,29]). Finally, using NFIRS special studies fields or new fields in modern systems such as the NERIS to capture additional details could aid our understanding of exposure

fires and SSD, as Appendix C identifies. However, these new fields would also increase the reporting burden of first responders, which is already challenging.

## 5. Conclusions

The NFIRS has provided a unique system to capture many pertinent details about exposure fires to aid in our understanding of these fires. Comprehensively documenting large (e.g., 2018 California Camp Fire) to moderately sized (e.g., 2011 Texas Tanglewood Fire) exposure fires might be impractical with the NFIRS, its modernization, the NERIS, or SIT-209. However, understanding locations where smaller exposure fires are common might identify areas of risk for large structure conflagrations as environmental conditions change. Also, comparing smaller-sized exposure fires in the NFIRS or NERIS to information from large exposure fires studied elsewhere (e.g., [11,29]) might aid our understanding of conditions that lead to large structures destroying exposure fires.

Using the NFIRS quantitatively to assess SSD's effect on structure-to-structure fire spread is challenging. However, the assessment presented here highlights that SSD does play some role in fire spread, mainly when containment of initial exposures does not occur. The high percentage of incidents with some defensive action that might alter heat fluxes highlights an often overlooked variable in assessing exposure fires. For example, the distance between surviving and destroyed homes is a significant predictor variable in WUI correlation studies [11,12] that have not considered the contributing role of defensive actions to the correlation between structure survivability and the distance to the nearest destroyed structures.

The NFIRS database does provide information on ignition pathways. As such, it could be used at select fires to understand ignition pathways better, mainly if additional data are collected. Also, the identified exposure pathways can aid in identifying general hazards and provide information to guide experimental efforts. However, using the NFIRS to quantify these pathways nationwide is challenging.

The NFIRS provides many attributes to help understand exposure fires. However, linking information on exposure fires with pre-fire, post-fire, and active-fire ground and aerial videos and information captured in NFIRS special study fields (Appendix C) or new fields in the NERIS can help advance our understanding of exposure fires and the SSD problem regarding fire spread from embers, flames, or radiant heat. Regardless, the lack of quantitative measures of heat fluxes, meteorological conditions, confounding factors such as defensive actions, and challenges with documenting basic exposure fire details curtails the ability to quantify the SSD problem with any exposure fire database.

Nevertheless, information from databases such as the NFIRS and NERIS could help guide well-designed experiments, facilitating a more quantitative understanding of how SSD contributes to fire spread. A database such as the NFIRS and others, even though limited in capturing the full extent of the exposure fire problem, can provide reality checks of experimental results while tracking the extent and conditions under which low SSD and other factors contribute to fire spread.

**Author Contributions:** Conceptualization, D.J.M. and W.E.M.; methodology, D.J.M.; software, D.J.M.; validation, D.J.M.; formal analysis, D.J.M.; investigation, D.J.M.; resources, D.J.M. and W.E.M.; data curation, D.J.M.; writing—original draft preparation, D.J.M.; writing—review and editing, W.E.M.; visualization, D.J.M.; supervision, W.E.M.; project administration, D.J.M. and W.E.M.; funding acquisition, W.E.M. All authors have read and agreed to the published version of the manuscript.

**Funding:** This research was funded in part by United States Forest Service (USFS) purchase orders 12045319P0025 and 1240BG21P0001. Purchase order 1240BG21P0001 utilized funding the California Department of Forestry and Fire Protection (CAL FIRE) provided and purchase order 12045319P0025 used funding from the USFS.

**Data Availability Statement:** All data utilized in this study are available through public sources.

**Acknowledgments:** We gratefully acknowledge Alexander Maranghides for providing feedback on an early manuscript version. We thank CAL FIRE and the U.S. Forest Service for funding this

effort. The U.S. Fire Administration staff is greatly appreciated for making NFIRS data publicly available. We thank the reviewers for their helpful suggestions. Finally, the dedicated fire protection professionals who contained the exposure fires and provided the source documentation for the NFIRS database are most appreciated.

**Conflicts of Interest:** The authors declare no conflicts of interest.

## Appendix A

The appendix contains the structured query language (SQL) statements used to extract exposure fires from the NFIRS database stored in a PostgreSQL database. Two attributes were added to each table in the PostgreSQL database. First, a primary key was added to each NFIRS table representing a concatenation of the state, fdid, inc_date, inc_no, and exp_no attributes, labeled incident_key. This attribute represented a unique value for each incident, enabling establishing the relationships between the BasicIncident, IncidentAddress, FireIncident tables [22].

Next, an attribute was added to the BasicIncident, IncidentAddress, FireIncident tables consisting of a concatenation of the state, fdid, inc_date, and inc_no attributes. This attribute was labeled inc_key_partial in the three tables. Exposure fires in the BasicIncident table were found by identifying records with more than one unique value for the inc_key_partial attribute.

1.  Dataset1 = SELECT BasicIncident.state, BasicIncident.fdid, BasicIncident.inc_date, BasicIncident.inc_no, BasicIncident.exp_no, BasicIncident.version, BasicIncident.dept_sta, BasicIncident.inc_type, BasicIncident.add_wild, BasicIncident.aid, BasicIncident.alarm, BasicIncident.arrival, BasicIncident.inc_cont, BasicIncident.lu_clear, BasicIncident.shift, BasicIncident.alarms, BasicIncident.district, BasicIncident.act_tak1, BasicIncident.act_tak3, BasicIncident.act_tak3, BasicIncident.app_mod, BasicIncident.sup_app, BasicIncident.ems_app, BasicIncident.oth_app, BasicIncident.sup_per, BasicIncident.ems_per, BasicIncident.oth_per, BasicIncident.resou_aid, BasicIncident.prop_loss, BasicIncident.cont_loss, BasicIncident.prop_val, BasicIncident.cont_val, BasicIncident.ff_death, BasicIncident.oth_death, BasicIncident.ff_inj, BasicIncident.oth_inj, BasicIncident.det_alert, BasicIncident.haz_rel, BasicIncident.mixed_use, BasicIncident.prop_use, BasicIncident.census, BasicIncident.incident_key, BasicIncident.inc_key_partial FROM BasicIncident WHERE BasicIncident.exp_no > 0;

2.  Dataset2 = SELECT BasicIncident.state, BasicIncident.fdid, BasicIncident.inc_date, BasicIncident.inc_no, BasicIncident.exp_no, BasicIncident.version, BasicIncident.dept_sta, BasicIncident.inc_type, BasicIncident.add_wild, BasicIncident.aid, BasicIncident.alarm, BasicIncident.arrival, BasicIncident.inc_cont, BasicIncident.lu_clear, BasicIncident.shift, BasicIncident.alarms, BasicIncident.district, BasicIncident.act_tak1, BasicIncident.act_tak3, BasicIncident.act_tak3, BasicIncident.app_mod, BasicIncident.sup_app, BasicIncident.ems_app, BasicIncident.oth_app, BasicIncident.sup_per, BasicIncident.ems_per, BasicIncident.oth_per, BasicIncident.resou_aid, BasicIncident.prop_loss, BasicIncident.cont_loss, BasicIncident.prop_val, BasicIncident.cont_val, BasicIncident.ff_death, BasicIncident.oth_death, BasicIncident.ff_inj, BasicIncident.oth_inj, BasicIncident.det_alert, BasicIncident.haz_rel, BasicIncident.mixed_use, BasicIncident.prop_use, BasicIncident.census, BasicIncident.incident_key, BasicIncident.inc_key_partial FROM BasicIncident JOIN Dataset1 ON BasicIncident.inc_key_partial = Dataset1.inc_key_partial;

3.  Dataset3 = SELECT DISTINCT Dataset2.state, Dataset2.fdid, Dataset2.inc_date, Dataset2.inc_no, Dataset2.exp_no, Dataset2.version, Dataset2.dept_sta, Dataset2.inc_type, Dataset2.add_wild, Dataset2.aid, Dataset2.alarm, Dataset2.arrival, Dataset2.inc_cont, Dataset2.lu_clear, Dataset2.shift, Dataset2.alarms, Dataset2.district, Dataset2.act_tak1, Dataset2.act_tak2, Dataset2.act_tak3, Dataset2.app_mod, Dataset2.sup_app, Dataset2.ems_app, Dataset2.oth_app, Dataset2.sup_per, Dataset2.ems_per, Dataset2.oth_per, Dataset2.resou_aid, Dataset2.prop_loss, Dataset2.cont_loss, Dataset2.prop_val, Dataset2.cont_val, Dataset2.ff_death, Dataset2.oth_death, Dataset2.ff_inj, Dataset2.

oth_inj, Dataset2.det_alert, Dataset2.haz_rel, Dataset2.mixed_use, Dataset2.prop_use, Dataset2.census, Dataset2.incident_key, .inc_key_partial FROM Dataset2;

4.  All_Exposures_Address = SELECT Dataset3.incident_key, Dataset3.state1, Dataset3. fdid, Dataset3.inc_date, Dataset3.inc_no, Dataset3.exp_no, Dataset3.version1, Dataset3. dept_sta, Dataset3.inc_type, Dataset3.add_wild, Dataset3.aid, Dataset3.alarm, Dataset3. arrival, Dataset3. inc_cont, Dataset3.lu_clear, Dataset3.shift, Dataset3.alarms, Dataset3. district, Dataset3.act_tak1, Dataset3.act_tak2, Dataset3.act_tak3, Dataset3.app_mod, Dataset3.sup_add, Dataset3.ems_app, Dataset3.oth_app, Dataset3.sup_per, Dataset3. ems_per, Dataset3.oth_per, Dataset3.resou_aid, Dataset3.prop_loss, Dataset3.cont_loss, Dataset3.prop_val, Dataset3.cont_val, Dataset3.ff_death, Dataset3.oth_death, Dataset3. ff_inj, Dataset3.oth_inj, Dataset3.det_alert, Dataset3.haz_rel, Dataset3.mixed_use, Dataset3.prop_use, Dataset3.census, Dataset3.inc_key_partial, IncidentAddress. incidentkey, IncidentAddress.loc_type, IncidentAddress.num_mile, IncidentAddress. street_pre, IncidentAddress.streetname, IncidentAddress.streettype, IncidentAddress. streetsuf, IncidentAddress.apt_no, IncidentAddress.city, IncidentAddress.state_id, IncidentAddress.zip5, IncidentAddress.zip4, IncidentAddress.x_street FROM Dataset3 JOIN IncidentAddress ON Dataset3.incident_key = IncidentAddress.incident_key;

5.  All_Exposure_Fires = Select * FROM All_Exposures_Addresses WHERE inc_type = 1 OR inc_type = 10 Or inc_type = 100 OR inc_type = 101 Or inc_type = 102 OR inc_type = 103 Or inc_type = 104 OR inc_type = 105 OR inc_type = 106 Or inc_type = 107 OR inc_type = 108 Or inc_type = 109 OR inc_type = 163 OR inc_type = 111 OR inc_type = 112 Or inc_type = 113 OR inc_type = 114 Or inc_type = 115 OR inc_type = 116 Or inc_type = 117 OR inc_type = 118 OR inc_type = 12 Or inc_type = 120 OR inc_type = 121 OR inc_type = 122 Or inc_type = 123 OR inc_type = 130 OR inc_type = 131 Or inc_type = 132 OR inc_type = 133 Or inc_type = 134 OR inc_type = 135 Or inc_type = 136 OR inc_type = 137 OR inc_type = 138 Or inc_type = 139 OR inc_type = 14 OR inc_type = 140 Or inc_type = 141 OR inc_type = 142 Or inc_type = 143 OR inc_type = 15 Or inc_type = 150 OR inc_type = 151 OR inc_type = 152 Or inc_type = 153 OR inc_type = 154 Or inc_type = 155 OR inc_type = 16 Or inc_type = 160 OR inc_type = 161 Or inc_type = 162 OR inc_type = 164 Or inc_type = 17 Or inc_type = 170 Or inc_type = 171 Or inc_type = 172 Or inc_type = 173;

6.  All_Exposure_Fires_No_Mutual_Aid = Select * FROM All_Exposure_Fires WHERE aid <> 3 or aid <> 4;

7.  All_Exposure_Fires_Inc_Type_Null = Select * FROM All_Exposure_Fires_No_Mutual_Aid WHERE inc_type Is Not Null.

Step 1 above provides a dataset of exposures where the exp_no is greater than 0, indicating the possibility of fires with more than one incident. Step 2 uses the query produced in step one to join back to the BasicIncident table, selecting all those exposures with the same values for the state, fdid, inc_date, and inc_no fields, thereby providing a dataset of all incidents with multiple exposures. The third step above joins the exposure fires from the BasicIncident table to the IncidentAddress table, providing address information for geolocation. The fourth step above selects only those representing fire incidents, as some exposures might be from non-fire incidents such as emergency medical service. Step five removes those incidents representing mutual aid exposures to avoid double-counting incidents. Finally, step six removes incidents where the inc_type field is null.

After examining the dataset from step seven above, we discovered that some incidents have an exp_no greater than zero but only contain a single incident or the queries above resulted in exposure fires with only one incident remaining. These fires are not exposure fires or representative of the complete exposure fire, and we removed these from the dataset using the queries below.

1.  All_Exposure_Fires_Inc_Type_Null_Count = Select COUNT(All_Exposure_Fires_Inc_Type_Null.inc_key_partial) As count_inc_key_partial FROM All_Exposures_Fires_Inc_Type_Null;

2.  All_Exposure)Fires_Finale = Select * FROM All_Exposures_Fires_Inc_Type_Null Join All_Exposure_Fires_Inc_Type_Null_Count ON All_Exposure_Fires_Inc_Type_Null. incident_key_partial = AllExposure_Fires_Inc_Type_Null_Count.incident_key_partial WHERE count_inc_key_partial is > 1.

We selected records in the FireIncident table with corresponding records in the filtered dataset above. The above steps provided the final exposure fire dataset from which we characterized NFIRS-reported exposure fires between 2002 and 2020. Additional filters were employed as detailed in the main document regarding the assessment of SSD.

**Appendix B**

This appendix contains tables showing all the NFIRS-reported exposure fires with significant incidents examined. Note that for the Tennesse Chimney Tops 2 Fire we included incidents that were coded as being from mutual aid. In this case, these incidents represent unique incidents and were not double counted.

**Table A1.** NFIRS-reported exposure fires with more than ten incidents showing affected (damaged or destroyed) buildings and total features affected compared to affected buildings from other sources.

| Fire Name and State | Date (Month-YY) | Buildings Affected (Sit-209 or Other Sources) | NFIRS Total Features Affected | NFIRS Total Buildings Affected |
| --- | --- | --- | --- | --- |
| TN Chimney Tops 2 Fire | Nov-16 | 2460 | 801 (Total 1884) | 798 |
| TN Chimney Tops 2 Fire | Nov-16 | 2460 | 73 (Total 1884) | 72 |
| TN Chimney Tops 2 Fire | Nov-16 | 2460 | 73 (Total 1884) | 72 |
| TN Black Bear Cub Fire | Mar-13 | 53 | 73 (Mutual Aid) | 73 |
| TX Tanglewood Fire | Feb-11 | 48 | 71 (Mutual Aid) | 69 |
| TN Chimney Tops 2 Fire | Nov-16 | 2460 | 508 (Total 1884) | 505 |
| TX Unknown | Apr-06 | Not Found | 20 (Mutual Aid) | 6 |
| CA Unknown | Nov-18 | Not Found | 17 (Mutual Aid) | 5 |
| TN Chimney Tops 2 Fire | Nov-16 | 2460 | 152 (Total 1884) | 151 |
| TN Chimney Tops 2 Fire | Nov-16 | 2460 | 146 (Total 1884) | 145 |
| TN Chimney Tops 2 Fire | Nov-16 | 2460 | 131 (Total 1884) | 130 |
| CA Witch Fire | Oct-07 | 1736 | 751 | 473 |
| CA Harris Fire | Oct-07 | 563 | 341 | 320 |
| AK McKinley Fire | Aug-19 | 84 | 325 | 127 |
| CA Clayton Fire | Aug-16 | 328 | 298 | 297 |
| AK Sockeye Fire | Jun-15 | 55 | 264 | 170 |
| CA Boles Fire | Sep-14 | 172 | 244 | 242 |
| CA Freeway Complex Fire | Nov-08 | 361 | 234 | 176 |
| CA Valley Fire | Sep-15 | 1280 | 221 | 218 |
| CA Rice Fire | Oct-07 | 248 | 209 | 208 |
| MN Ham Lake | May-07 | 133 | 164 | 163 |
| OK Unknown | Dec-05 | Not Found | 151 | 20 |
| CA Trabing Fire | Jun-08 | 20 | 147 | 93 |
| CA Humboldt Fire | Jun-08 | 261 | 142 | 117 |
| CA Summit Fire | May-08 | 99 | 139 | 86 |
| CA Telegraph Fire | Jul-08 | 39 | 134 | 131 |

**Table A1.** *Cont.*

| Fire Name and State | Date (Month-YY) | Buildings Affected (Sit-209 or Other Sources) | NFIRS Total Features Affected | NFIRS Total Buildings Affected |
|---|---|---|---|---|
| FL Leigh Fire | Apr-06 | Not Found | 127 | 28 |
| CA BTU Lightning Complex Fire | Aug-08 | 117 | 123 | 119 |
| AR Chaffee Fire | Jan-08 | 150 | 122 | 121 |
| NM Quail Ridge Fire | Feb-11 | 15 | 101 | 17 |
| CA Clover Fire | Sep-13 | 211 | 91 | 59 |
| TX Unknown | Jan-06 | Not Found | 90 | 21 |
| OK Harrah Fire | Mar-11 | 39 | 85 | 22 |
| KS Starbuck Fire (Northwest Oklahoma Complex) | Mar-17 | Not Found | 78 | 74 |
| FL 36 Lincoln and E. 6th Street Fire | May-13 | 3 | 73 | 14 |
| FL Leigh Fire | Apr-06 | Not Found | 73 | 14 |
| KS Highlands Fire (Northwest Oklahoma Complex) | Mar-17 | Not Found | 68 | 17 |
| WI Germann Road Fire | May-13 | 47 | 66 | 66 |
| KS Starbuck Fire (Northwest Oklahoma Complex) | Mar-17 | Not Found | 62 | 58 |
| TX Willow Creek Fire | Feb-11 | 29 | 59 | 39 |
| SC Windsor Greens Fire | Mar-13 | 118 | 57 | 26 |
| OK Unknown | Dec-05 | Not Found | 54 | 1 |
| CA Courtney Fire | Sep-14 | 69 | 53 | 51 |
| CA Cocos Fire | May-14 | 40 | 52 | 51 |
| TX Ringgold Texas Fire | Jan-06 | 50 | 50 | 49 |
| KS Unknown | Mar-06 | Not Found | 46 | 35 |
| IA Unknown | Mar-05 | Not Found | 44 | 6 |
| KS Unknown | Mar-16 | Not Found | 43 | 41 |
| CA Round Fire | Feb-15 | 40 | 42 | 41 |
| NV Caughlin Fire | Nov-11 | 29 | 41 | 40 |
| CA Unknown | Dec-20 | Not Found | 41 | 33 |
| MN Green Valley Fire | May-13 | 58 | 40 | 39 |
| TX Unknown | Mar-06 | Not Found | 38 | 14 |
| MI Unknown | Nov-05 | Not Found | 35 | 0 |
| CA Stagecoach Fire | Aug-20 | 60 | 34 | 18 |
| AR Chaffee Fire | Jan-08 | 150 | 34 | 34 |
| FL Unknown | Mar-09 | Not Found | 33 | 6 |
| CO Coal Seam Fire | Jun-02 | 44 | 31 | 17 |
| CA Unknown | May-06 | Not Found | 31 | 3 |
| WY Unknown | Apr-02 | Not Found | 29 | 15 |
| CA Ophir Fire | Jun-08 | 49 | 28 | 25 |
| ID Sweetwater Fire | Aug-08 | 21 | 27 | 26 |

**Table A1.** *Cont.*

| Fire Name and State | Date (Month-YY) | Buildings Affected (Sit-209 or Other Sources) | NFIRS Total Features Affected | NFIRS Total Buildings Affected |
|---|---|---|---|---|
| FL Unknown | Mar-07 | Not Found | 27 | 5 |
| CO Unknown | Jun-12 | Not Found | 27 | 26 |
| MI Unknown | May-18 | Not Found | 26 | 0 |
| CA Bully Fire | Jul-14 | 20 | 25 | 14 |
| NE Valentine Fire | Jul-06 | 42 | 24 | 22 |
| TX Unknown | Dec-05 | Not Found | 24 | 7 |
| CA Unknown | Sep-07 | Not Found | 24 | 10 |
| GA Sweat Farm Again Fire | Jun-11 | 14 | 23 | 17 |
| GA Unknown | Apr-07 | Not Found | 23 | 23 |
| TX Unknown | Nov-05 | Not Found | 22 | 14 |
| FL Unknown | Mar-07 | Not Found | 22 | 8 |
| TX Unknown | Apr-11 | Not Found | 21 | 15 |
| OK Unknown | Aug-11 | Not Found | 21 | 7 |
| TX Pitt Road Fire | May-11 | 16 | 20 | 10 |
| CA Unknown | May-17 | Not Found | 20 | 19 |
| CA Lockheed Fire | Aug-09 | 14 | 19 | 12 |
| MT Roaring Lion Fire | Jul-16 | 16 | 19 | 18 |
| WA Boffer Fire | Aug-18 | 9 | 17 | 11 |
| CA Poomacha Fire | Oct-07 | 217 | 16 | 15 |
| FL Unknown | Feb-17 | Not Found | 16 | 12 |
| OH Unknown | Sep-10 | Not Found | 14 | 2 |
| SC Unknown | Apr-14 | Not Found | 14 | 12 |
| CA Vail Fire | Sep-09 | 15 | 11 | 5 |
| GA Unknown | Mar-07 | Not Found | 11 | 10 |
| SC Unknown | Mar-18 | Not Found | 11 | 11 |
| CA Washoe Fire | Aug-07 | 6 | 10 | 8 |
| CA Unknown | Jul-20 | Not Found | 10 | 9 |

**Table A2.** List of other incident types (beyond buildings) for the NFIRS-reported exposure fires listed in Table A1.

| Fire Name and State | NFIRS Confined Structure | NFIRS Veg. | NFIRS Mobile Struct. | NFIRS Other | NFIRS Outside | NFIRS Other than Building | NFIRS Not Defined Structure | NFIRS Vehicle |
|---|---|---|---|---|---|---|---|---|
| TN Chimney Tops 2 Fire | 0 | 3 | 0 | 0 | 0 | 0 | 0 | 0 |
| TN Chimney Tops 2 Fire | 0 | 1 | 0 | 0 | 0 | 0 | 0 | 0 |
| TN Chimney Tops 2 Fire | 0 | 1 | 0 | 0 | 0 | 0 | 0 | 0 |
| TN Black Bear Cub Fire | 0 | 0 | 0 | 0 | 0 | 0 | 0 | 0 |
| TX Tanglewood Fire | 0 | 1 | 0 | 0 | 0 | 0 | 0 | 1 |
| TN Chimney Tops 2 Fire | 0 | 3 | 0 | 0 | 0 | 0 | 0 | 0 |
| TX Unknown | 1 | 1 | 1 | 0 | 0 | 0 | 0 | 10 |

**Table A2.** *Cont.*

| Fire Name and State | NFIRS Confined Structure | NFIRS Veg. | NFIRS Mobile Struct. | NFIRS Other | NFIRS Outside | NFIRS Other than Building | NFIRS Not Defined Structure | NFIRS Vehicle |
|---|---|---|---|---|---|---|---|---|
| CA Unknown | 1 | 7 | 0 | 0 | 4 | 0 | 0 | 0 |
| TN Chimney Tops 2 Fire | 0 | 1 | 0 | 0 | 0 | 0 | 0 | 0 |
| TN Chimney Tops 2 Fire | 0 | 1 | 0 | 0 | 0 | 0 | 0 | 0 |
| TN Chimney Tops 2 Fire | 0 | 1 | 0 | 0 | 0 | 0 | 0 | 0 |
| CA Witch Fire | 0 | 1 | 0 | 273 | 4 | 0 | 0 | 0 |
| CA Harris Fire | 0 | 1 | 0 | 0 | 0 | 20 | 0 | 0 |
| AK McKinley Fire | 0 | 1 | 21 | 0 | 0 | 3 | 0 | 173 |
| CA Clayton Fire | 0 | 1 | 0 | 0 | 0 | 0 | 0 | 0 |
| AK Sockeye Fire | 0 | 1 | 15 | 0 | 8 | 23 | 0 | 47 |
| CA Boles Fire | 0 | 1 | 1 | 0 | 0 | 0 | 0 | 0 |
| CA Freeway Complex Fire | 0 | 4 | 6 | 14 | 1 | 15 | 0 | 18 |
| CA Valley Fire | 0 | 0 | 3 | 0 | 0 | 0 | 0 | 0 |
| CA Rice Fire | 0 | 1 | 0 | 0 | 0 | 0 | 0 | 0 |
| MN Ham Lake | 0 | 1 | 0 | 0 | 0 | 0 | 0 | 0 |
| OK Unknown | 0 | 59 | 0 | 49 | 0 | 0 | 0 | 23 |
| CA Trabing Fire | 0 | 1 | 0 | 2 | 0 | 0 | 0 | 51 |
| CA Humboldt Fire | 0 | 1 | 12 | 4 | 0 | 2 | 0 | 6 |
| CA Summit Fire | 0 | 1 | 0 | 8 | 0 | 0 | 0 | 44 |
| CA Telegraph Fire | 0 | 1 | 1 | 1 | 0 | 0 | 0 | 0 |
| FL Leigh Fire | 0 | 1 | 0 | 67 | 0 | 1 | 0 | 30 |
| CA BTU Lightning Complex Fire | 0 | 1 | 2 | 0 | 0 | 1 | 0 | 0 |
| AR Chaffee Fire | 0 | 1 | 0 | 0 | 0 | 0 | 0 | 0 |
| NM Quail Ridge Fire | 0 | 1 | 0 | 0 | 43 | 0 | 0 | 40 |
| CA Clover Fire | 0 | 1 | 31 | 0 | 0 | 0 | 0 | 0 |
| TX Unknown | 0 | 11 | 4 | 33 | 1 | 0 | 0 | 20 |
| OK Harrah Fire | 0 | 33 | 2 | 2 | 0 | 26 | 0 | 0 |
| KS Starbuck Fire (Northwest Oklahoma Complex) | 0 | 1 | 0 | 0 | 0 | 3 | 0 | 0 |
| FL 36 Lincoln and E. 6th Street Fire | 0 | 2 | 0 | 4 | 3 | 13 | 0 | 37 |
| FL Leigh Fire | 0 | 2 | 0 | 4 | 3 | 13 | 0 | 37 |
| KS Highlands Fire (Northwest Oklahoma Complex) | 0 | 40 | 0 | 0 | 1 | 1 | 0 | 9 |
| WI Germann Road Fire | 0 | 0 | 0 | 0 | 0 | 0 | 0 | 0 |
| KS Starbuck Fire (Northwest Oklahoma Complex) | 0 | 1 | 0 | 0 | 0 | 3 | 0 | 0 |
| TX Willow Creek Fire | 0 | 15 | 2 | 0 | 1 | 0 | 0 | 2 |
| SC Windsor Greens Fire | 0 | 0 | 0 | 0 | 0 | 0 | 0 | 31 |

**Table A2.** *Cont.*

| Fire Name and State | NFIRS Confined Structure | NFIRS Veg. | NFIRS Mobile Struct. | NFIRS Other | NFIRS Outside | NFIRS Other than Building | NFIRS Not Defined Structure | NFIRS Vehicle |
|---|---|---|---|---|---|---|---|---|
| OK Unknown | 0 | 50 | 0 | 0 | 0 | 0 | 0 | 3 |
| CA Courtney Fire | 0 | 1 | 0 | 1 | 0 | 0 | 0 | 0 |
| CA Cocos Fire | 0 | 1 | 0 | 0 | 0 | 0 | 0 | 0 |
| TX Ringgold Texas Fire | 0 | 1 | 0 | 0 | 0 | 0 | 0 | 0 |
| KS Unknown | 0 | 2 | 0 | 0 | 8 | 0 | 0 | 1 |
| IA Unknown | 0 | 38 | 0 | 0 | 0 | 0 | 0 | 0 |
| KS Unknown | 0 | 1 | 1 | 0 | 0 | 0 | 0 | 0 |
| CA Round Fire | 0 | 1 | 0 | 0 | 0 | 0 | 0 | 0 |
| NV Caughlin Fire | 0 | 1 | 0 | 0 | 0 | 0 | 0 | 0 |
| CA Unknown | 0 | 1 | 6 | 0 | 0 | 0 | 0 | 1 |
| MN Green Valley Fire | 0 | 1 | 0 | 0 | 0 | 0 | 0 | 0 |
| TX Unknown | 0 | 5 | 11 | 0 | 0 | 0 | 0 | 8 |
| MI Unknown | 0 | 1 | 0 | 0 | 1 | 0 | 0 | 33 |
| CA Stagecoach Fire | 0 | 0 | 9 | 0 | 0 | 7 | 0 | 0 |
| AR Chaffee Fire | 0 | 0 | 0 | 0 | 0 | 0 | 0 | 0 |
| FL Unknown | 0 | 16 | 3 | 0 | 0 | 2 | 0 | 6 |
| CO Coal Seam Fire | 0 | 0 | 14 | 0 | 0 | 0 | 0 | 0 |
| CA Unknown | 0 | 6 | 1 | 1 | 3 | 7 | 0 | 10 |
| WY Unknown | 0 | 2 | 1 | 0 | 0 | 1 | 0 | 10 |
| CA Ophir Fire | 0 | 2 | 1 | 0 | 0 | 0 | 0 | 0 |
| ID Sweetwater Fire | 0 | 1 | 0 | 0 | 0 | 0 | 0 | 0 |
| FL Unknown | 0 | 1 | 0 | 16 | 0 | 0 | 0 | 5 |
| CO Unknown | 0 | 1 | 0 | 0 | 0 | 0 | 0 | 0 |
| MI Unknown | 0 | 25 | 0 | 0 | 0 | 0 | 0 | 1 |
| CA Bully Fire | 0 | 1 | 4 | 0 | 0 | 0 | 0 | 6 |
| NE Valentine Fire | 0 | 1 | 1 | 0 | 0 | 0 | 0 | 0 |
| TX Unknown | 0 | 7 | 0 | 0 | 9 | 1 | 0 | 0 |
| CA Unknown | 0 | 1 | 1 | 10 | 0 | 1 | 0 | 1 |
| GA Sweat Farm Again Fire | 0 | 1 | 0 | 0 | 0 | 0 | 0 | 5 |
| GA Unknown | 0 | 0 | 0 | 0 | 0 | 0 | 0 | 0 |
| TX Unknown | 0 | 1 | 2 | 0 | 0 | 0 | 0 | 5 |
| FL Unknown | 0 | 1 | 7 | 0 | 1 | 0 | 0 | 5 |
| TX Unknown | 0 | 1 | 0 | 0 | 0 | 0 | 0 | 5 |
| OK Unknown | 0 | 1 | 6 | 0 | 0 | 6 | 0 | 1 |
| TX Pitt Road Fire | 0 | 0 | 2 | 6 | 0 | 0 | 0 | 2 |
| CA Unknown | 0 | 1 | 0 | 0 | 0 | 0 | 0 | 0 |
| CA Lockheed Fire | 0 | 1 | 1 | 1 | 0 | 1 | 0 | 3 |
| MT Roaring Lion Fire | 0 | 1 | 0 | 0 | 0 | 0 | 0 | 0 |

**Table A2.** *Cont.*

| Fire Name and State | NFIRS Confined Structure | NFIRS Veg. | NFIRS Mobile Struct. | NFIRS Other | NFIRS Outside | NFIRS Other than Building | NFIRS Not Defined Structure | NFIRS Vehicle |
|---|---|---|---|---|---|---|---|---|
| WA Boffer Fire | 0 | 0 | 0 | 0 | 6 | 0 | 0 | 0 |
| CA Poomacha Fire | 0 | 1 | 0 | 0 | 0 | 0 | 0 | 0 |
| FL Unknown | 0 | 1 | 3 | 0 | 0 | 0 | 0 | 0 |
| OH Unknown | 0 | 9 | 0 | 0 | 3 | 0 | 0 | 0 |
| SC Unknown | 0 | 1 | 0 | 0 | 0 | 1 | 0 | 0 |
| CA Vail Fire | 0 | 2 | 2 | 0 | 1 | 0 | 0 | 1 |
| GA Unknown | 0 | 1 | 0 | 0 | 0 | 0 | 0 | 0 |
| SC Unknown | 0 | 0 | 0 | 0 | 0 | 0 | 0 | 0 |
| CA Washoe Fire | 0 | 0 | 0 | 0 | 0 | 0 | 0 | 2 |
| CA Unknown | 0 | 1 | 0 | 0 | 0 | 0 | 0 | 0 |

**Appendix C**

This appendix describes a list of NFIRS "special study" fields or fields to potentially be included in the NFIRS modernization, the National Emergency Response Information System (NERIS), that might be utilized to capture information to understand better the role of SSD and ignition pathways in fire spread at exposure fires. These fields are in addition to the existing fields in the NFIRS database, including alarm times, incident types, heat sources, defensive actions, structure types, property uses, the life and property losses associated with the fire, and other attributes already contained in NFIRS that result in NFIRS supporting a robust method to document exposure fires if implemented as designed. However, collecting information at moderately to large exposure fires like the large WUI fires examined here is challenging across significant extents.

Also, most databases of exposure fires do not currently associates active-fire images and videos with the incidents. These images and videos, when recorded, can be found on social media or other sources on the internet or are part of the first responders' apparatus and have been used to understand defensive actions at large WUI fires. Also, first responders and structure owners present during fires sometimes record videos and images of active fire conditions. Integrating these media, eyewitness accounts, and other information at the national level, individual states, or even select fires can help advance our understanding of exposure fires. Finally, the ability to link exposure fire data with post-fire aerial and ground imagery in systems such as the NFIRS and NERIS will aid in advancing our understanding of conditions leading to damage and destruction at exposure fires.

**Table A3.** Additional fields beyond those contained in the NFIRS to better capture information at exposure fires. Note that some of these fields can be captured through NFIRS special study fields or existing fields in the NFIRS that might not always be used. Additionally, systems in development, such as the NERIS, might include these fields in their database schemas.

| Special Study Field | Description |
|---|---|
| Link to Other Databases | This field stores the fire's unique identifier from other databases such as SIT-209. |
| Geolocation Information | This value would be represented as two fields containing an X and Y coordinate (e.g., latitude and longitude) representing the precise location of the burned feature. The centroid can be used for larger features, such as wildland fires. |

**Table A3.** *Cont.*

| Special Study Field | Description |
| --- | --- |
| Specific Minor Feature | This field would expand on the *inc_type* field in the NFIRS, explicitly identifying those incidents categorized as structures other than buildings (e.g., fences). |
| Abandoned | This field would expand on the NFIRS *prop_use* field, explicitly identifying those incidents occurring on properties that have been abandoned. |
| Record of Igniting Feature | This field would store a unique identifier of the record of the incident that ignited the incident in this record. This field would allow for documenting the order of ignitions if known. |
| Flame Length | This field would describe the lengths of the flames produced by the incident. |
| Flame Angle | This field would describe the angle of the flame produced by the incident. |
| Internal or External Ignition | This field would describe if the fire spread through an internal (e.g., broken window or vent) or external to the structure (see feature ignited below) ignition. |
| Feature Ignited | This field would describe the specific feature ignited in a single/multifamily exposure fire if the ignition were external. A list of values would be provided, such as eaves, deck, window frame, wood shake roof, and other pertinent values. Ideally materials and characteristics of the feature would be documented. |
| Broken Window | This field would describe if the internal ignition was due to a broken window. |
| Window Type | This field would describe the type of window (e.g., double-pane or single-pane, and possibly other values) if the ignition due to an internal ignition and the failure of the window. |
| Vent Ignition | This field would describe if the internal ignition was due to ember entry through a vent. |
| Vent Type | This field would describe the type of vent if the ignition were due to an internal ignition through the vent. |
| Link to Videos and Images | This field could provide links to videos and images of the incident if available. |
| Field Measured SSD | This field would contain the field-measured structure or feature separation distance. |
| Remarks | This field would record any remarks by the authority having jurisdiction regarding SSD and ignition pathways. |

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
