# Peer review of "Examining Exposure Fires from the United States National Fire Incident Reporting System between 2002 and 2020"

_fire, doi:10.3390/fire7030074_

Round 1

Reviewer 1 Report

Comments and Suggestions for Authors

This work presented an extensive assessment of exposure fires from the NFIRS database, including the economic costs, affected feature types, and property utilization patterns for these exposure fires. It also examined structure separation distance. This work provides a good statistical analysis on historic exposure fires, which is the base for further study.

Some minor comments:

1.       Line78, what is “NFRIA”

2.       Line 359, note the format

Author Response

Thank you so much for the review. We made all the corrections you requested.

Reviewer 2 Report

Comments and Suggestions for Authors

The work is interesting, the method and conclusion is clear.My suggestions are given as follow:

1、For the scenario construction of exposure fires, some  influcening factors ase essential, such as categories of initial fires, propogation criterion, categories of secondary fires. Quantitative conclusion should be achieved to add the innovation.

Comments on the Quality of English Language

It's easy to read.

Reviewer 3 Report

Comments and Suggestions for Authors

This manuscript analyses US fires reported between 2002 and 2020 for both WUI and non-WUI systems.  The authors used NFIRS data for their analysis. This paper represents interesting results, which are important since the frequency and intensity of fires in the US have increased in the past few decades due to fuel accumulation/management and climate change.  For example, the authors analyzed the correlation between building survival and distance, which is a critical parameter for fire prediction.

The manuscript is well written. I have only two major and a few minor comments:

Major comment

1.     Very limited literature was discussed in the Introduction section (only 8). It would be helpful if the authors present some recent research, e.g.,

Radeloff, V. C., Mockrin, M. H., Helmers, D., Carlson, A., Hawbaker, T. J., Martinuzzi, S., Schug, F., Alexandre, P. M., Kramer, H. A., & Pidgeon, A. M. (2023). Rising wildfire risk to houses in the United States, especially in grasslands and shrublands. Science, 382(6671), 702-707.

Also, some web pages were accessed a few years ago. It would be good if they are visited recently.

2.     The intro is also missing a big picture (with references) of why studying WUI and non-WUI fires is important especially because of warmer temperatures, fuel accumulation, changed precipitation patterns, etc. 

Some minor comments

In the text. “Figure 1. “1” is in Italic and it should be corrected (and for the rest of the figures as well).

Line 109. Reference on Grass Fire (and other fires in the text) are missing (e.g., refs on CalFare web and others).

Line 230. This sentence is not needed.

Fig. 2 (and others). The numbers front is too small.

Lines 350-360 – font and paragraph alignment should be fixed.

Line 930. The year in the reference is wrong.

The references should be checked and fixed.

In summary, I recommend this manuscript to be published after fixing 2 major and a few minor revisions.

Comments on the Quality of English Language

Minor
